# Learning to Respond:
# A Large-Scale Benchmark and Progressive Learning Framework for Trigger-Centric Online Video Understanding

## Abstract

The rapid growth of online video platforms resulted in vast amounts of streaming and surveillance content, creating an urgent demand for real-time video understanding. Unlike offline tasks, online video understanding emphasizes proactive responsiveness, where models must detect when sufficient evidence has appeared in the stream to answer a given question (*trigger*) and respond immediately. However, current studies provide insufficient exploration of such capabilities. To bridge this gap, we introduce TV-Online (Trigger Video-Online), a large-scale dataset with $50K$ videos, $200K$ questions, and $500K$ time-stamped answers. TV-Online covers progressively complex trigger-based tasks, ranging from basic temporal grounding to asynchronous scheduling and multi-trigger reasoning. These tasks motivate an agent-like modeling paradigm in which the system continuously processes streaming inputs and decides at each step whether to respond or remain silent. We instantiate this paradigm with a streaming-oriented model that employs protocol-level tagging and structured state management to regulate frame-by-frame decisions, ensuring precise response timing and consistent handling of asynchronous, multi-turn triggers. To endow the model with such capabilities, we adopt a progressive training strategy that leverages difficulty annotations in TV-Online and reinforcement objectives to shape responsiveness, coverage, and coherence across evolving interactions. Finally, we introduce a unified evaluation metric that integrates semantic, temporal, and coverage dimensions to holistically assess online video understanding. Extensive experiments demonstrate that TV-Online, together with the proposed model, training strategy, and metric, provides a comprehensive benchmark for advancing trigger-oriented online video understanding toward practical real-time video intelligence.

## 1 Introduction

The rapid expansion of online video platforms has led to an unprecedented proliferation of multimedia content, spanning diverse formats such as live streaming and surveillance video(Ansari et al., 2023; Kumar et al., 2024; Qian et al., 2024; Zhou et al., 2024). Effectively interpreting and analyzing such content in real time has become increasingly critical(S et al., 2025; Prawiro et al., 2020; Shidik et al., 2019; Qian et al., 2024). This capability enables intelligent systems to extract meaningful insights from complex, dynamic, and multi-modal visual environments(Zhou et al., 2024; Chen et al., 2024; 2025a; Qian et al., 2025; Yang et al., 2025c; Han et al., 2025; Wang et al., 2025c). Consequently, advances in online video understanding hold broad potential for applications in domains such as robotics, autonomous driving, and live broadcasting, establishing this area as an emerging and rapidly growing field of research(Chen et al., 2024; 2025a; Qian et al., 2024).

Recent advances in multimodal large models (MLLMs) provide new opportunities for online video understanding.(Qian et al., 2024; Chen et al., 2024; 2025a; Qian et al., 2025; Yang et al., 2025c; Han et al., 2025) Nevertheless, compared with offline tasks, online video understanding introduces more stringent requirements, including real-time processing, long-range contextual reasoning, and proactive responses.(Zhou et al., 2024; Wang et al., 2025c) At the core of such responsiveness lies the

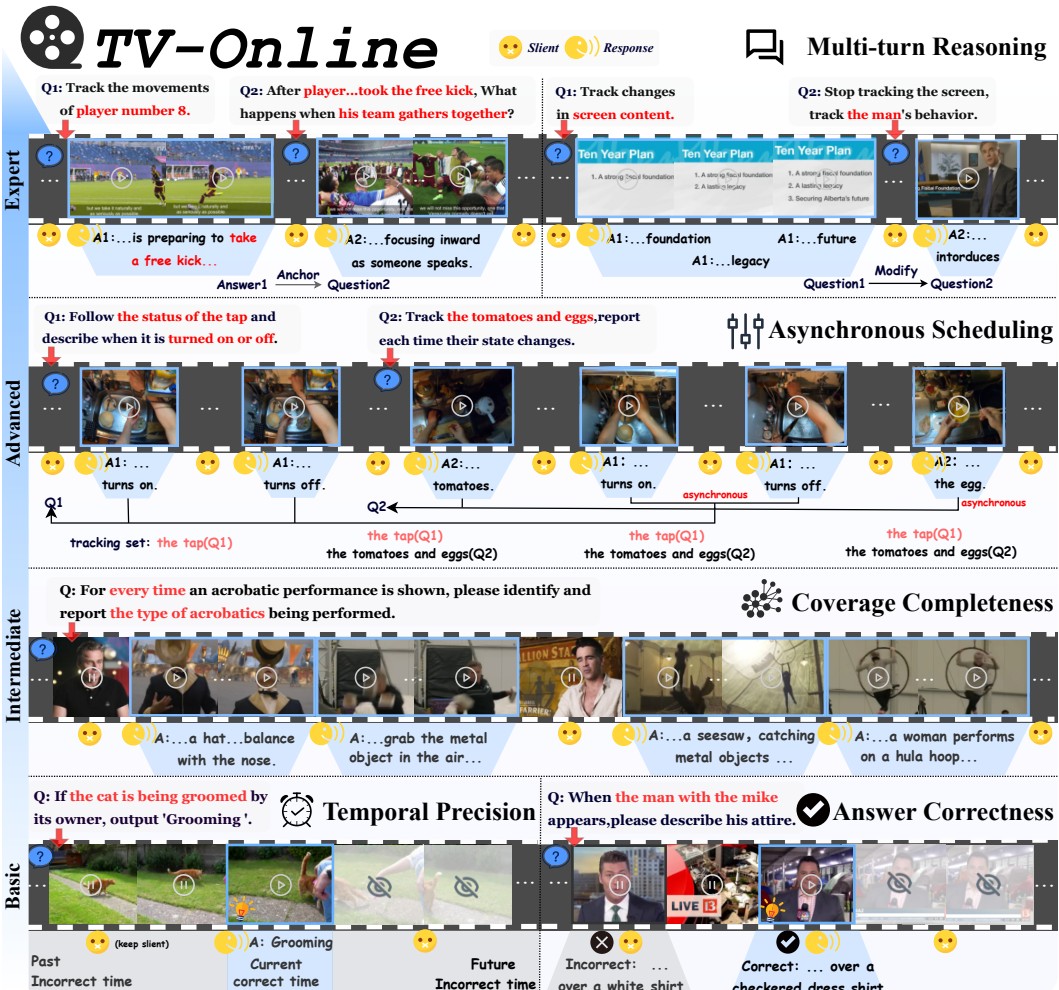

**Figure 1: Overview of the TV-Online dataset.** The dataset is structured hierarchically, starting with temporal precision and answer correctness at the base layer. The next layer adds coverage completeness, requiring detection of all relevant answers. On top of this, asynchronous scheduling introduces simultaneous tracking of multiple independent triggers. Finally, multi-turn reasoning demands handling sequential triggers with dynamic context updates. This layered structure enables systematic evaluation from basic grounding to advanced reasoning.

notion of a *trigger*, defined as a temporal or semantic cue that connects a query to the corresponding event, action, or state within a video(Yang et al., 2025c; Han et al., 2025; Huang et al., 2025b; Niu et al., 2025). The trigger mechanism enables systems to move from passive perception to active decision-making(Chen et al., 2024; 2025a; Qian et al., 2025; Yang et al., 2025c).

However, the exploration of triggers remains limited. As illustrated in Figure 1,our dataset introduces a hierarchy of challenges under the frame-level streaming setting. At the basic layer, **Temporal Precision** and **Answer Correctness** establish the foundational capabilities, requiring models to accurately identify the timing of events (e.g., grooming at the current moment) and provide correct answers (e.g., checked dress shirt), ensuring reliable grounding in both time and content. The intermediate layer, **Coverage Completeness**, demands comprehensive event detection across time, such as identifying all acrobatic actions in a sequence, requiring robust and continuous tracking. Above this, **Asynchronous Scheduling** introduces the challenge of multi-object, independent tracking (e.g., simultaneously monitoring a tap and tomatoes/eggs), necessitating the management of multiple evolving triggers. At the Expert layer, **Multi-turn Reasoning** involves handling complex, sequential triggers with dynamic context updates (e.g., tracking player movements and team behavior), relying on memory, inference, and adaptation to evolving contexts. These challenges concern temporal precision, answer correctness, and coverage completeness across diverse events.

At more advanced levels, they require handling asynchronous scheduling under densely distributed triggers and performing multi-turn reasoning with cross-trigger memory and semantic coherence.

While prior work has highlighted the significance of trigger-based responsiveness for online video comprehension, existing efforts fall short in addressing real-world complexity. However, they still exhibit limitations in addressing more practical real-world scenarios: 1) Existing benchmarks rarely capture time-evolving and progressively triggered queries, leaving a gap with the inherently dynamic nature of online video streams(Yang et al., 2025c; Huang et al., 2025b; Niu et al., 2025). 2) Most approaches remain restricted to single-trigger scenarios, offering limited insights into richer, multi-trigger dynamics(Qian et al., 2024; Chen et al., 2024; Zhou et al., 2024). 3) Previous evaluation metrics focus primarily on answer accuracy while neglecting temporal sensitivity, which is essential for assessing online video understanding(Qian et al., 2024; Zhou et al., 2024; Yang et al., 2025c; Huang et al., 2025b; Niu et al., 2025; Han et al., 2025; Wang et al., 2025c).

These limitations underscore the need for a trigger-oriented benchmark. To this end, we introduce **TV-Online** (**T**rigger **V**ideo-**Online**), a large-scale dataset comprising $50K$ videos, $200K$ questions, and $500K$ time-stamped answers. TV-Online is organized into three hierarchical levels: (i) single-answer triggers for basic temporal localization and online semantic comprehension, (ii) multi-answer triggers for comprehensive coverage of temporally distributed events, and (iii) inter-dependent triggers for modeling complex semantic and temporal relations across multiple queries and answers. Building on this hierarchy, we formulate online video understanding from an agent perspective. At each time step, the model (**TOM** Trigger-Oriented Model) receives streaming frames and user queries, and must decide whether to respond or remain silent. To support this agent-style formulation, we introduce lightweight mechanisms for explicit response control, real-time inference, and state management to ensure precise trigger–answer alignment. On top of this design, we adopt a progressive training pipeline, where the model first learns fundamental online comprehension and multi-trigger handling, and is then optimized through reinforcement learning to improve temporal accuracy, coverage completeness, and cross-turn consistency. To further support fair and holistic evaluation, we introduce a dedicated metric that jointly accounts for temporal precision, semantic correctness, and coverage completeness. Our contributions are summarized as follows:

- We introduce *TV-online*, the first large-scale dataset explicitly designed for trigger-oriented online video understanding, comprising $50K$ video samples, $200K$ questions, and $500K$ time-stamped answers across three hierarchical trigger levels, together with a dedicated evaluation metric tailored to our dataset.

- We develop an agent-style framework *TOM* that unifies response control, efficient inference, and state management, and adopt a progressive training strategy with reinforcement learning to enhance temporal accuracy, coverage completeness, and cross-turn consistency.

- We conduct extensive experiments demonstrating our model's state-of-the-art performance on TV-Online, establishing a strong baseline for advancing online video understanding.

## 2 RELATED WORK

### 2.1 VIDEO ONLINE DATASETS

With the advancement of multimodal large language models (MLLMs), increasing attention has been directed towards the construction of datasets for online video understanding (Yang et al., 2025c; Han et al., 2025; Huang et al., 2025b; Xun et al., 2024; Lin et al., 2024b; Niu et al., 2025; Wang et al., 2025c; Wu et al., 2025). SVBench Yang et al. (2025c), Shot2Story Han et al. (2025), OVBench Huang et al. (2025b), and RTV-Bench Xun et al. (2024) advance the field by adopting structured cross-segment QA and temporal reasoning paradigms to probe long-context temporal understanding and fine-grained narrative comprehension. StreamingBench Lin et al. (2024b), SteamBench Wu et al. (2025) and Live CC series Chen et al. (2025a) focus on real-time comprehension and interactive reasoning, yet they are still constrained by reliance on pre-recorded content. OVO-Bench Niu et al. (2025) and OmniMMI-v01 Wang et al. (2025c) employ purpose-built interactive datasets designed for proactive engagement, incorporating explicit mechanisms to facilitate active user interaction. However, current benchmarks still lack strong online triggers and multi-turn interrelated queries, thereby offering only a partial representation of trigger-specific properties (Yang et al., 2025c; Han et al., 2025; Huang et al., 2025b; Xun et al., 2024; Niu et al., 2025; Wang et al., 2025c;b).

## 2.2 VIDEO ONLINE METHODS

With the rapid proliferation of real-time video applications, online video understanding has garnered increasing research attention due to its potential to support timely, context-aware responses in continuous and untrimmed video streams (Liu et al., 2023; Qian et al., 2024; Zhou et al., 2024). Early efforts in this domain primarily focused on adapting offline video recognition and captioning models to streaming scenarios by introducing constraints on computation and memory usage (Liu et al., 2023; Di et al., 2025; Yang et al., 2025b; Li et al., 2025a; Yao et al., 2025; Qian et al., 2024; Zhou et al., 2024). A prominent line of subsequent research has concentrated on minimizing latency and memory overhead in long video stream processing through techniques such as cache compression, memory selection, and Differential Token Drop (Liu et al., 2023; Di et al., 2025; Yang et al., 2025b; Li et al., 2025a; Yao et al., 2025; Qian et al., 2024; Zhou et al., 2024; Yao et al., 2025; Huang et al., 2025b; Xiong et al., 2025). VideoLLM-onlineChen et al. (2024) and DispiderQian et al. (2025) focus on developing MLLMs specifically designed for streaming video input and long-context reasoning. Agent-based approaches, such as StreamAgent Yang et al. (2025a), introduce proactive planning to balance immediate responses with deferred information acquisition. Nevertheless, existing methods still face limitations in realistic, trigger-driven scenarios. To address these challenges, we introduce *TV-online*, a unified framework that spans from single-trigger and one-to-many cases to complex multi-trigger scenarios. It categorizes triggers by difficulty, evaluates performance along temporal–semantic–coverage axes, and adopts a progressive training pipeline to systematically enhance reasoning across increasing levels of trigger complexity.

**Table 1: TV-Online vs. existing online video datasets**. TV-Online is the first large-scale dataset designed for trigger-based online video understanding. It integrates three categories of data that encapsulate the most practically essential capabilities required in trigger-oriented scenarios. "Proactive (cnt)": The number of QA tasks that require a proactive, trigger-based response.

| Benchmark | Videos | Questions | Answers | Avg. Turns | Open | Total Len. | Online | Multi-turn | Proactive (cnt) |
|---|---|---|---|---|---|---|---|---|---|
| OVOBench(Niu et al., 2025) | 644 | 2,814 | 2,814 | - | - | 76.73 h | ✓ | ✗ | ✓ (172) |
| Videochat-Online(Huang et al., 2025b) | 1,463 | 7,000 | 7,000 | - | - | 45.93 h | ✗ | ✗ | ✗ (0) |
| RTV-Bench(Xun et al., 2024) | 552 | 4,608 | 4,608 | - | - | 167.2 h | ✗ | ✗ | ✗ (0) |
| StreamingBench(Lin et al., 2024b) | 900 | 4,500 | 4,500 | - | - | 60.78 h | ✓ | ✗ | ✓ (50) |
| SVBench(Yang et al., 2025c) | 1,353 | 49,979 | 49,979 | 4.29 | 49,979 | 54.5 h | ✗ | ✓ | ✗ (0) |
| OmniMMI(Wang et al., 2025c) | 1,121 | 2,290 | 2,290 | 2.62 | 2,290 | 100.98 h | ✓ | ✓ | ✓ (1,072) |
| Shot2Story(Han et al., 2025) | 42,958 | 11,370 | 11,370 | - | 11,370 | 204.25 h | ✗ | ✗ | ✗ (0) |
| StreamBench(Xiong et al., 2025) | 306 | 1,800 | 1,800 | 5.99 | 1,800 | 25 h | ✓ | ✓ | ✗ (0) |
| VideoLLM-Online(Chen et al., 2024) | 698 | 64,246 | 338,236 | 3.05 | 64,246 | 68.44 h | ✓ | ✓ | ✓ (16,500) |
| MMDuetIT(Wang et al., 2024) | 52,767 | 76,343 | 118,496 | 1.55 | 39,291 | 618.90 h | ✓ | ✗ | ✓ (109,000) |
| Timechat Online(Yao et al., 2025) | 11,043 | 139,000 | 139,000 | - | - | 1,000.86 h | ✗ | ✗ | ✗ (0) |
| Ours(TV-Online) | 51,444 | 267,581 | 542,603 | 2.03 | 267,581 | 1,502.18 h | ✓ | ✓ | ✓ (267,581) |

## 3 DATASET

### 3.1 PROBLEM FORMULATION

Within online video understanding, the trigger refers to a specific cue within the video that prompts the model to respond to a given query. Each trigger encompasses two essential dimensions: a temporal component, indicating the timestamp at which the query is raised in the video stream, and a semantic component, which specifies the concrete content of the query. Trigger-based tasks require models to satisfy two strict criteria: (i) the generated response must be semantically aligned with the query, and (ii) the predicted timestamp must accurately correspond to the relevant video segment. Unlike conventional settings, the model is expected to produce answers only once sufficient evidence appears in the video stream. Building on this definition, our TV-Online organizes trigger-based tasks into three progressive categories, each designed to assess increasingly levels of model capability.

### 3.2 TASK DEFINITION

**Temporal Grounded Question Answering (TGQA)**: This task is defined as the core and most elementary trigger-based task in online video understanding. In this task, the model is required to generate an answer immediately once sufficient evidence has appeared in the video stream. The

**Figure 2:** We introduce a structured, multi-stage pipeline to synthesize our hierarchically structured dataset, TV-online, from a wide range of heterogeneous sources. To ensure the quality and integrity of this data, our pipeline incorporates a tailored, two-stage validation process. This process includes type-specific QA checks adapted to the nuances of each source, and a dedicated timestamp verification strategy to guarantee high temporal accuracy.

objective is to evaluate the model's fundamental trigger capability, which entails simultaneously achieving precise temporal grounding and semantic correctness. Formally, TGQA is defined as requiring the model to accurately identify the appropriate triggering point within the video and to generate a response in strict accordance with the given instruction. To reflect different dimensions of perceptual and reasoning demands, TGQA is further organized into four subtypes: Visual Perception, Information Integration, Status Confirmation and Reasoning.

**Temporal State Tracking (TST)**: This task requires the model to output the complete set of answers associated with a given trigger throughout the video stream. Since a single trigger in live streaming may correspond to multiple distributed answers. This task challenges the model's ability to capture all relevant answers based on the fundamental capabilities specified in TGQA. From the dimensions of entity quantity, state, and attribute in video streams, we define three subtasks: Numerical Attribute Tracking, State Attribute Tracking and Set Tracking. These subtypes evaluate a model's ability to monitor quantitative changes, capture qualitative state transitions, and maintain temporal tracking. This setting highlights the challenge of handling evolving answers within a single trigger.

**Sequential Query Understanding (SQU)**: In streaming scenarios, multiple queries may arise at different time points. SQU is designed to accommodate this requirement by allowing triggers to be raised at arbitrary moments during the video stream. The model must determine the appropriate trigger to respond to at each moment and model inter-trigger dependencies and contextual relationships. At the same time, it should consider whether historical context and prior responses need to be incorporated for integrated reasoning. This ensures that the response is both appropriate to the current query and consistent within the ongoing dialogue. We define four sub-tasks under SQU: Cancel, Transfer, Sequential-Forward, and Parallel. Specifically, Parallel evaluates the ability to bind asynchronously triggered queries to their corresponding answers. Cancel and Transfer probe inter-query relationship modeling. Sequential-Forward examines the use of historical responses for reasoning. Together, these sub-tasks form a structured benchmark that targets key capabilities for managing asynchronous, multi-turn interactions in sequential query understanding.

### 3.3 DATASET CONSTRUCTION

We construct TV-Online by integrating multiple video corpora, including Shot2Story, Ego4D, Vript, DiDeMo, Charades (Han et al., 2025; Grauman et al., 2022; Yang et al., 2024; Anne Hendricks et al., 2017; Sigurdsson et al., 2016). These sources span diverse annotation styles: dense segment-level labelingHan et al. (2025); Yang et al. (2024), fine-grained frame-level annotations Grauman et al. (2022), and spatio-temporal localization with precise event boundaries (Anne Hendricks et al., 2017; Sigurdsson et al., 2016). To enable trigger-based online video understanding, we construct a systematic pipeline that leverages LLM-based data synthesis to unify heterogeneous annotations into standardized QA pairs. To ensure reliability, we design a two-stage validation process addressing both QA quality and temporal accuracy as shown in Figure2.

### 3.4 DATASET ANALYSIS

Our TV-Online dataset contains $50K$ videos, $200K$ questions, and $500K$ time-stamped answers. Our dataset consists of three main categories and 11 subcategories, covering various challenges

posed by online trigger-based scenarios. The distribution of video categories is illustrated in Table 2. The details of our TV-Online can be found in the Appendix C.

**Table 2: Data Distribution**. Videos and QA (Q/A) per subcategory with Overall totals.

| Category | Status Confirmation | Visual Perception | Information Integration | Reasoning | State Attribute Tracking | Numerical Attribute Tracking |
|---|---|---|---|---|---|---|
| **Videos** | 38,681 | 27,078 | 13,025 | 856 | 21,855 | 30273 |
| **QA** | 45k/45k | 37k/37k | 15k/15k | 1k/1k | 41k/127k | 32k/78k |
| **Category** | Set Tracking | Parallel | Transfer | Sequential Forward | Cancel | Overall |
| **Videos** | 4,956 | 21,071 | 6,552 | 10,479 | 3,686 | 51,441 |
| **QA** | 5k/16k | 36k/102k | 25k/65k | 22k/56k | 8k/0k | 267k/542k |

## 4 METHOD

We formulate Trigger-online video understanding from the agent perspective. At each time step, the model takes streaming frames and user queries as input and performs a decision process on whether to generate a response or remain silent. To support this setting, our model introduces a set of special tags to regulate streaming behavior and a fast inference mechanism to ensure real-time efficacy. In addition, a queue-based state management module maintains temporal context and enables precise trigger–answer alignment. Based on this design, we adopt a progressive training pipeline. The first two stages pre-train and fine-tune the model for fundamental online comprehension and multi-trigger handling. The final post-training stage leverages reinforcement learning with task-specific rewards to optimize temporal accuracy, coverage completeness, and cross-turn consistency.

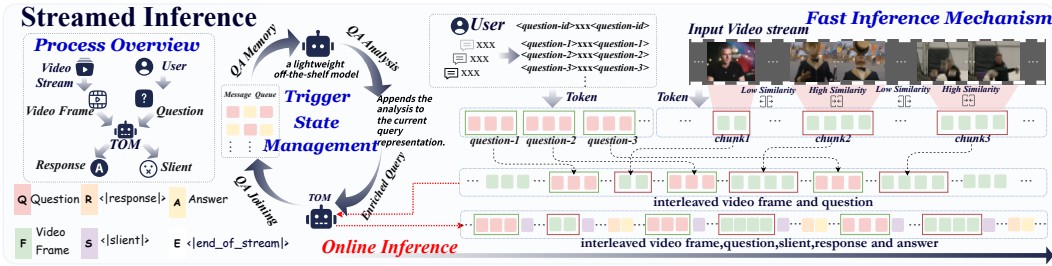

**Figure 3:** Our proposed method incorporates queue-based state management and fast inference to process streaming video and queries in real time, ensuring precise trigger–answer alignment.

### 4.1 MODEL

As shown in Figure3, we introduce a set of special tokens to regulate streaming behavior: $response$ / $silent$ control whether the model outputs a response or remains silent at each frame, $EOS$ marks the termination of the video stream, while $question_{id}$ and $answer_{id}$ assign unique identifiers to queries and bind each response to its corresponding query. These tags collectively form a lightweight protocol that standardizes the interaction between streaming inputs and model outputs.

To meet real-time constraints, we introduce **Fast Inference Mechanism**. When consecutive frames exhibit high visual similarity, the model is encouraged to skip redundant processing by directly outputting $silent$, thereby accelerating streaming inference without sacrificing temporal precision. Please refer to Appendix B.1 for the detailed algorithm and threshold ablation.

To better adapt the model to multi-trigger scenarios, we propose **Trigger State Management**. Specifically, we maintain a structured queue to record queries and corresponding answers. Upon the arrival of a new trigger, we leverage a lightweight off-the-shelf model to analyze its relevance to historical triggers and previously generated answers. This process identifies potential dependencies, contextual links, or contradictions. The derived relational information is appended to the current query representation. The enriched query is then forwarded to the main model, enabling it to make context-aware predictions that remain consistent across multiple triggers and interactions. Please refer to Appendix B.1 for the architectural details and examples of the query enrichment process.

## 4.2 TRAINING PIPELINE

During training, we adopt a fully streaming paradigm, where the video is fed frame by frame and the model is required to analyze each incoming frame in sequence. Specifically, we employ a two-stage pretraining scheme, which constrains the model to operate on continuously arriving information and ensures that its behavior aligns with the characteristics of real-world online video understanding.

In Stage I, we leverage offline video datasets to pre-train our model. The query is inserted at the beginning of each video. The model is required to remain silent during non-informative spans and to produce event descriptions at the end of each event segment. The model is supervised to remain silent on non-informative spans and to respond near pseudo-event boundaries. To mitigate boundary bias from scene segmentation, we introduce temporal jitter and near-boundary hard negatives. This stage equips the model with the fundamental ability to operate under a streaming paradigm, teaching it when to remain silent and when to respond, thereby establishing the basic online behavior.

Stage II fine-tunes the model on TV-Online, where questions appear at arbitrary positions in the video stream, requiring dynamic responses as the context unfolds. To enhance performance in complex online scenarios, we enforce explicit trigger-answer linking and coverage-aware objectives. Additionally, the data is augmented with special tags to indicate such associations, enabling the model to learn this mechanism and better handle asynchronous triggers and concurrent queries.

## 4.3 POST TRAINING

To align the model with the challenges of our settings, we introduce a post-training stage that incorporates reinforcement learning with task-specific reward functions. First in scenarios where a single trigger is associated with multiple valid answers, we employ a recall-oriented reward to promote coverage of all valid responses. Complementarily, a precision-oriented reward is introduced to penalize redundant or spurious outputs:

$$R_{\mathrm{r}}(O^{\mathrm{pred}}, O^{\mathrm{gt}}, q_i) = \frac{|TP|}{|O^{\mathrm{gt}}|} \quad R_{\mathrm{p}}(O^{\mathrm{pred}}, O^{\mathrm{gt}}, q_i) = \frac{|TP|}{|O^{\mathrm{pred}}|} \quad R_{F_\beta} = \frac{(1+\beta^2)R_p * R_r}{\beta^2 * R_p + R_r} \quad (1)$$

$$TP = \left\{ o_t^{\mathrm{pred}} \;\middle|\; \exists\, o_{t'}^{\mathrm{gt}} \in O^{\mathrm{gt}}, t' \in [t - \tfrac{W}{2},\, t + \tfrac{W}{2}], \mathrm{Eval}(o_t^{\mathrm{pred}}, o_{t'}^{\mathrm{gt}}|q_i) = 1,\, t' \notin \mathcal{M} \right\} \quad (2)$$

where $O^{\mathrm{pred}}$ is the sequence of model predictions, $O^{\mathrm{gt}}$ is the sequence of ground-truth answers, $W$ is the temporal alignment window size, $\mathrm{Eval}(\cdot, \cdot \mid q_i)$ is a binary LLM evaluator conditioned on the query $q_i$, and $\mathcal{M}$ denotes the set of ground-truth indices already matched to previous predictions.

Second, in multi-trigger settings, concurrent triggers may occur within overlapping temporal windows, making it difficult to bind answers to the correct queries. To resolve this, we introduce *Tag Reward* that explicitly links triggers and answers, and we reward the model for correct tag usage. To discourage spurious tags that do not correspond to any ground-truth trigger, we introduce *Hallucination Penalty* proportional to the fraction of unmatched tags:

$$R_{tag} = \underbrace{\frac{1}{T} \sum_{t=1}^{T} \mathbf{1}\Big( o_t^{\mathrm{pred}} \supset \texttt{<answer-k>} \; k \in \mathcal{K}^{\mathrm{gt}} \Big)}_{\text{Tag Reward}} - \underbrace{\left( \lambda \cdot \frac{\left| \{ \texttt{<answer-k>} \in O^{\mathrm{pred}} : k \notin \mathcal{K}^{\mathrm{gt}} \} \right|}{|O^{\mathrm{pred}}|} \right)}_{\text{Hallucination Penalty}}$$

$$(3)$$

where $T$ is the total number of prediction steps, $o_t^{\mathrm{pred}}$ denotes the prediction at step $t$, $\{o_1^{\mathrm{pred}}, \ldots, o_T^{\mathrm{pred}}\}$ is the set of all predictions, $\mathcal{K}^{\mathrm{gt}}$ is the set of ground-truth answer indices, $\mathbf{1}(\cdot)$ is the indicator function returning 1 if the condition holds and 0 otherwise, and $\lambda \in [0, 1]$ is a hyper-parameter controlling the penalty strength. Final reward is the aggregation of $R = \alpha \cdot R_{F_\beta} + \gamma \cdot R_{\mathrm{tag}}$, where $\alpha$ and $\gamma$ are used to adjust the weights of the $R_{F_\beta}$ and tag-based reward.

This mechanism encourages accurate referencing and updating of previous answers, ensuring coherence across multi-turn triggers. Together, these rewards refine the model's decision-making, ensuring precise timing, coverage, and consistency in asynchronous scenarios. Please refer to Appendix B.6 for additional training details.

## 5 EXPERIMENTS

### 5.1 EVALUATION METRIC

Existing metrics fail to capture the unique requirements of online video understanding, where answers must be not only semantically correct but also temporally aligned and complete. As illustrated in Figure 4. We design a task-specific evaluation metric along three dimensions: **semantic accuracy**, **coverage completeness**, and **temporal latency**. Formally, given ground truth $GT = \{(A_i, t_i)\}_{i=1}^{N_{GT}}$ and predictions $Pred = \{(P_j, t'_j)\}_{j=1}^{N_{Pred}}$, a prediction is considered correct if it falls within a tolerance window $W$ of the ground-truth timestamp and is semantically consistent. We compute evaluation metrics using standard F1 definitions.

### 5.2 IMPLEMENTATION DETAILS

Our model is built on the Qwen2.5-VL-3B Bai et al. (2025) backbone and implemented with PyTorch and VERL. Training is conducted on 8×A800 GPUs. Videos are sampled at 1 fps, with each frame represented by 48 visual tokens. The training follows a three-stage strategy: pretraining on timestamped datasets (Shot2Story and Vript), then instruction fine-tuning and reinforcement learning on TV-Online. The entire process is optimized with AdamW and completes in about 36 hours.

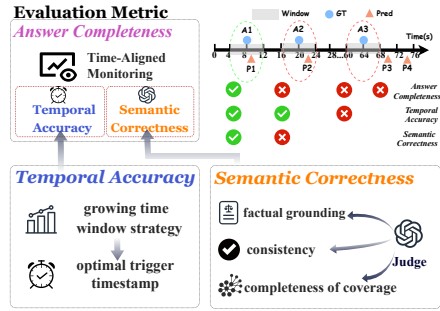

**Figure 4:** Illustration of the proposed evaluation metric.

**Table 3:** Main results of the our proposed model (TOM) and various baselines on the TV-online benchmark. It can be seen that our method achieves the state-of-art performance

| Model | Frames | Param | W | TGQA | | | TST | | | SQU | | | Overall | | |
|---|---|---|---|---|---|---|---|---|---|---|---|---|---|---|---|
| | | | | Pre | Recall | F1-Score | Pre | Recall | F1-Score | Pre | Recall | F1-Score | Pre | Recall | F1-Score |
| Offline and Streaming Models (*without proactive output capability*) | | | | | | | | | | | | | | | |
| GPT-4o(OpenAI et al., 2024) | 16 | / | – | 67.4 | 67.4 | 67.4 | 42.1 | 42.1 | 42.1 | 34.5 | 34.5 | 34.5 | 46.0 | 46.0 | 46.0 |
| Qwen2.5-VL(Bai et al., 2025) | 16 | 3B | – | 60.2 | 60.2 | 60.2 | 30.7 | 30.7 | 30.7 | 24.8 | 24.8 | 24.8 | 36.1 | 36.1 | 36.1 |
| Qwen2.5-VL(Bai et al., 2025) | 16 | 7B | – | 63.7 | 63.7 | 63.7 | 33.5 | 33.5 | 33.5 | 27.8 | 27.8 | 27.8 | 38.4 | 38.4 | 38.4 |
| MiniCPM-V-4.5(Yao et al., 2024) | 16 | 8B | – | 62.6 | 62.6 | 62.6 | 32.6 | 32.6 | 32.6 | 26.2 | 26.2 | 26.2 | 37.6 | 37.6 | 37.6 |
| GLM-4.1V(Team et al., 2025) | 16 | 9B | – | 64.3 | 64.3 | 64.3 | 38.5 | 38.5 | 38.5 | 28.5 | 28.5 | 28.5 | 40.7 | 40.7 | 40.7 |
| IntenVL-2.5(Chen et al., 2025b) | 16 | 2B | – | 49.3 | 49.3 | 49.3 | 26.9 | 26.9 | 26.9 | 20.9 | 20.9 | 20.9 | 27.4 | 27.4 | 27.4 |
| Online Models (*with proactive output capability*) | | | | | | | | | | | | | | | |
| VideoLLM-online(Chen et al., 2024) | 1fps | 7B | 4s | 2.1 | 3.5 | 2.6 | 5.6 | 6.7 | 6.1 | 1.9 | 2.4 | 2.1 | 3.2 | 4.2 | 3.6 |
| LiveCC(Chen et al., 2025a) | 1fps | 7B | 4s | 2.5 | 3.6 | 3.0 | 6.6 | 7.8 | 7.2 | 1.4 | 2.8 | 1.9 | 3.7 | 4.8 | 4.2 |
| TimeChat-online(Yao et al., 2025) | 1fps | 7B | 4s | 9.87 | 20.2 | 13.3 | 17.2 | 17.9 | 17.5 | 13.6 | 10.8 | 12.0 | 13.3 | 15.1 | 14.1 |
| TOM (Ours) | 1fps | 3B | 4s | 51.6 | 40.3 | 45.2 | 36.2 | 33.8 | 35.0 | 36.7 | 28.0 | 31.8 | 42.2 | 33.8 | 37.5 |

### 5.3 MAIN RESULTS

Since most MLLMs lack native support for online streaming and proactive response generation, we design two evaluation settings for fair and comprehensive comparison. In the online setting, the model first receives all historical context and then streaming frames at 1 fps, where it must autonomously decide when to respond; temporal accuracy is assessed within a tolerance window $W$. In the offline setting, the model is given a short clip centered on the ground-truth timestamp for each answer, testing semantic understanding at the critical moment without requiring temporal reasoning. We provide a detailed discussion on these two evaluation settings in Appendix E.1.

As shown in Table3, our proposed TOM surpasses all online baselines and most offline models of equivalent parameter scale, even when the latter are provided ground-truth trigger times, demonstrating the effectiveness of our holistic approach, which we further validate through ablation studies.

### 5.4 ABLATION STUDY

To evaluate the effectiveness of each component, we conduct a comprehensive ablation study. Results show that each training stage contributes cumulatively to higher Overall F1 (Table 4), validating

**Table 4:** Ablation Study of Training Stages

| Stage 1 | Stage 2 | Stage 3 | Overall F1 (%) |
|---------|---------|---------|----------------|
|         | ✓       |         | 32.6           |
| ✓       | ✓       |         | 34.2 (+1.6)    |
|         | ✓       | ✓       | 36.4 (+3.8)    |
| ✓       | ✓       | ✓       | **37.5** (+4.9) |

**Table 5:** Trigger State Management(TSM)

| Method | SQU (%) | Overall (%) |
|--------|---------|-------------|
| TOM (w/o TSM) | 28.5 | 36.3 |
| **TOM (w/ TSM)** | **31.8** | **37.5** |
| **Performance Gain** | **+3.3** | **+1.2** |

**Table 6:** TOM vs Videollm-online

| Method | Overall (%) |
|--------|-------------|
| Videollm-online | 27.5 |
| **TOM(Ours)** | **32.6** |
| **Performance Gain** | **+5.1** |

**Table 7:** Question/Answer ID Tag Protocal.

| Method | TGQA | TST | SQU | Overall |
|--------|------|-----|-----|---------|
| TOM (w/o ID Tags) | 44.1 | 31.2 | 20.0 | 30.7 |
| **TOM (w/ ID Tags)** | **45.2** | **35.0** | **31.8** | **37.5** |
| **Performance Gain** | **+1.0** | **+3.8** | **+11.8** | **+6.8** |

**Table 8:** Parameter $\beta$ of reward function

| Reward ($F_\beta$) | P (%) | R (%) | F1 (%) | Silent (%) |
|--------------------|-------|-------|--------|------------|
| Recall-Biased ($\beta = 2.0$) | 40.7 | **34.5** | 37.3 | 78.1 |
| Precision-Biased ($\beta = 0.5$) | **43.3** | 31.0 | 36.1 | 88.3 |
| **Balanced (Ours, $\beta = 1.0$)** | 42.2 | 33.8 | **37.5** | 82.4 |

our progressive strategy. Incorporating the Decoupled Dialogue Management Module (DMM) improves SQU by +3.3% and Overall by +1.2% (Table 5), while the ID tagging mechanism brings large gains, including +11.8 on SQU and +6.8 overall (Table 7). Varying the parameter $\beta$ in the $F_\beta$ reward (Table 8) shows that a balanced setting ($\beta = 1.0$) yields the best trade-off between performance and silent rate. We also compare our method with VideoLLM-Online(Chen et al., 2024). We adapt VideoLLM-Online to the TV-Online training setup and train it under the same conditions. The results, shown in Table 6, demonstrate the superiority of our approach. Additionally, we provide supplementary ablation studies in the Appendix, including investigations into inference constraints (FPS and token budgets in Appendix D.2) and model capacity scaling (3B vs. 7B in Appendix D.3), which further verify the robustness and scalability of our framework.

## 5.5 GENERALIZATION ON OTHER BENCHMARKS

To demonstrate the robust generalization capability of our framework, we extend our evaluation to **ProactiveVideoQA** Wang et al. (2025b), **StreamingBench** Lin et al. (2024b), **OVO-Bench** Niu et al. (2025), and **ViSpeak-Bench** (Fu et al., 2025b). As shown in Tables 9, 10, and 11, ToM-7B consistently achieves superior performance. Specifically, it attains a state-of-the-art overall accuracy of **37.35** on ProactiveVideoQA, a leading score of **48.78** on Streaming-Bench, and **54.68%** on OVO-Bench's active responding task. On ViSpeak-Bench, our model secures the second-best overall performance, demonstrating strong competitiveness against existing baselines. Furthermore, additional evaluations on **OmniMMI** Wang et al. (2025c) and **SVBench** Yang et al. (2025c) are provided in Appendix D.1, where ToM achieves state-of-the-art results on online subtasks and notably outperforms the closed-source Gemini-1.5-Pro on SVBench.

## 5.6 CASE STUDY

As illustrated in Figure5. At $t = 5$s, the model detects a shot, and at $t = 9$s responds "The blue player has scored." At $t = 15$s, when asked about the follow-up action, the model leverages its trigger state to infer that two players are celebrating. Red boxes mark key frames, blue question marks denote queries, lightbulbs indicate responses, and the right panel shows trigger state tracking.

## 5.7 DISCUSSION

To the best of our knowledge, this is the first work that frames online video understanding as an agent making streaming decisions in a dynamic environment. Rather than passively

**Figure 5: Case Study of TOM**. TOM answering queries in a football match by detecting key events, maintaining trigger states, and producing coherent multi-turn responses.

mapping inputs to outputs, the model is organized into perception (temporal representation), reasoning (trigger–answer alignment), and action (responding or remaining silent). From this perspective,

even simple structural designs (e.g. such as *Trigger State Management* Table 5 and *Tag Protocol* Table 7) significantly enhance performance in multi-trigger and asynchronous scenarios. Furthermore, reinforcement learning introduces two complementary rewards: one favoring accuracy, which makes the agent more conservative and inclined to remain silent. The other favoring completeness, which makes it more proactive and responsive. This behavioral trade-off shows that reward signals not only improve performance but also shape the agent's decision-making style. In this way, our framework not only boosts performance but also offers an adjustable solution for online understanding that more closely resembles real agent behavior.

**Table 9:** Results on ProactiveVideoQA. $\omega$ denotes the temporal tolerance window; different values indicate varying degrees of strictness for response timing. Our proposed TOM method achieves state-of-the-art results.

| Method | Frames | WEB | | | EGO | | | TV | | | VAD | | | Overall |
|---|---|---|---|---|---|---|---|---|---|---|---|---|---|---|
| | | $\omega=0$ | $\omega=0.5$ | $\omega=1$ | $\omega=0$ | $\omega=0.5$ | $\omega=1$ | $\omega=0$ | $\omega=0.5$ | $\omega=1$ | $\omega=0$ | $\omega=0.5$ | $\omega=1$ | |
| VideoLLM-Online*(Chen et al., 2024) | – | 25.9 | 25.9 | 25.9 | 25.0 | 25.0 | 25.1 | 17.8 | 18.3 | 18.8 | 25.0 | 25.0 | 25.0 | 23.7 |
| MMDuet*(Wang et al., 2024) | – | 37.2 | 38.9 | 40.7 | **44.0** | **46.0** | **47.9** | 20.7 | 21.1 | 21.6 | **26.4** | **27.4** | **28.5** | 34.68 |
| ToM-3B | 64 | 39.2 | 44.0 | 48.9 | 31.1 | 33.3 | 35.6 | 24.4 | 24.5 | 24.6 | 22.3 | 21.9 | 21.5 | 32.65 |
| ToM-7B (ours) | 64 | **40.1** | **45.1** | **50.1** | 36.1 | 39.5 | 42.9 | **27.1** | **28.2** | **29.3** | 26.2 | 26.6 | 27.1 | **37.35** |

**Table 10:** Results on StreamingBench and OvO-Bench. Our proposed TOM method outperforms all baseline models on both the overall score of StreamingBench and the forward active responding score of OVO-Bench, demonstrating its superior overall performance.

| Method | Frames | StreamingBench | | | | OVO-Bench Forward Active Responding |
|---|---|---|---|---|---|---|
| | | Real-Time Visual | Omni-Source | Contextual | Overall | |
| VideoLLM-Online 8B(Chen et al., 2024) | 2 fps | 35.99 | 28.45 | 26.55 | 30.33 | – |
| FlashVStream 7B(Zhang et al., 2024a) | 1 fps | 23.23 | 26.00 | 24.12 | 24.45 | 44.23 |
| Dispider 7B(Qian et al., 2025) | 1 fps | 67.63 | 35.66 | 33.61 | 45.63 | 34.72 |
| ViSpeak 7B (s2)(Fu et al., 2025b) | 1 fps | 74.36 | – | – | – | 54.25 |
| TimeChat-Online(Yao et al., 2025) | 1 fps | 75.36 | – | – | – | 36.70 |
| StreamBridge(Wang et al., 2025a) | 1 fps | 77.04 | – | – | – | 48.36 |
| StreamForest(Zeng et al., 2025) | 1 fps | **77.26** | – | – | – | 53.49 |
| Qwen2.5-VL(Bai et al., 2025) | 1/0.5/0.2 fps | 73.76 | 35.20 | 29.30 | 46.08 | 40.27 |
| ToM-7B (ours) | 1/0.5/0.2 fps | 74.72 | **37.30** | **34.20** | **48.78** | **54.68** |

**Table 11:** Results on ViSpeak-Bench. Time Accuracy and Text Score are reported for each event category, as well as the overall averages. Our proposed ToM method achieves the second-best performance.

| Method | Params | Frames | Time Accuracy (%) | | | | | | | Text Score | | | | | | | | Overall |
|---|---|---|---|---|---|---|---|---|---|---|---|---|---|---|---|---|---|---|
| | | | AW | VI | HR | VW | VT | GU | All | VR | AW | VI | HR | VW | VT | GU | All | |
| InternVL-2.5*(Chen et al., 2025b) | 8B | 16 | 41.50 | 55.00 | 46.00 | 96.00 | 72.00 | 99.50 | 68.42 | 2.93 | 2.16 | 3.67 | 4.81 | 1.26 | 2.66 | | 1.98 |
| Qwen2.5-VL*(Bai et al., 2025) | 72B | 1 fps | 44.50 | 81.00 | 77.00 | 91.00 | 91.00 | 93.00 | 79.58 | 3.15 | 2.64 | 3.36 | 1.00 | 5.00 | 5.00 | 1.50 | 3.09 | 2.62 |
| VITA 1.5*(Fu et al., 2025a) | 7B | 1 fps | 18.00 | 46.00 | 40.00 | 88.00 | 49.00 | 97.50 | 56.42 | 2.40 | 2.08 | 0.57 | 0.85 | 4.57 | 4.49 | 1.18 | 2.31 | 1.54 |
| Ola*(Liu et al., 2025) | 7B | 1 fps | 27.00 | 67.00 | 44.00 | 89.00 | 69.00 | 98.50 | 65.75 | 2.95 | 1.81 | 2.67 | 0.55 | 4.71 | 1.52 | 2.55 | | 1.86 |
| FlashVstream*(Zhang et al., 2024a) | 7B | 1 fps | 34.00 | 16.00 | 48.00 | 75.00 | 33.00 | 99.50 | 50.92 | 1.75 | 1.63 | 1.31 | 0.67 | 4.88 | 4.61 | 0.70 | 2.22 | 1.24 |
| Dispider*(Qian et al., 2025) | 7B | 16 | 38.50 | 70.00 | 44.00 | 69.00 | 100.00 | 99.50 | 70.17 | 2.50 | 1.75 | 4.06 | 0.91 | 0.61 | 2.49 | 2.07 | 2.06 | 1.63 |
| Qwen2.5-VL(Bai et al., 2025) | 7B | 1 fps | 33.50 | 77.00 | 35.00 | 95.00 | 85.00 | 100.00 | 70.91 | 2.17 | 1.97 | 3.80 | 0.85 | 5.00 | 3.80 | 1.13 | 2.75 | 2.16 |
| ViSpeak (s3)*(Fu et al., 2025b) | 7B | 1 fps | 56.50 | 72.00 | 83.00 | 93.00 | 79.00 | 99.00 | **80.42** | 3.75 | 2.63 | 3.84 | 1.07 | 4.95 | 3.15 | 3.36 | **3.25** | **2.76** |
| ToM-7B (ours) | 7B | 1 fps | 44.50 | 80.00 | 50.00 | 95.00 | 83.00 | 99.00 | 75.00 | 2.34 | 1.89 | 4.45 | 0.86 | 4.95 | 3.64 | 1.25 | 2.76 | 2.30 |

## 6 CONCLUSION

In this work, we introduce TV-Online, a large-scale benchmark for trigger-oriented online video understanding, together with TOM (Trigger-Oriented Model), a streaming framework equipped with protocol-level tagging and dialogue management. We further propose a progressive three-stage training strategy and a unified evaluation metric integrating temporal accuracy, semantic correctness, and coverage completeness. Extensive experiments and ablation studies demonstrate that our approach achieves state-of-the-art performance and provides a solid baseline for future research. We believe this work lays the foundation for practical real-time video intelligence, and future directions include scaling to larger multimodal large language model backbones, incorporating richer contextual memory, and to broader online reasoning tasks.

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

## A  IMPACT STATEMENTS

### A.1  ETHICS STATEMENT

This work adheres to the ICLR Code of Ethics. In this study, no human subjects or animal experimentation was involved. All datasets used, including TV-Online, were sourced in compliance with relevant usage guidelines, ensuring no violation of privacy. We have taken care to avoid any biases or discriminatory outcomes in our research process. No personally identifiable information was used, and no experiments were conducted that could raise privacy or security concerns. We are committed to maintaining transparency and integrity throughout the research process.

### A.2  REPRODUCIBILITY STATEMENT

We have made every effort to ensure that the results presented in this paper are reproducible. All code and datasets have been made publicly available in an anonymous repository to facilitate replication and verification. The experimental setup, including training steps, model configurations, and hardware details, is described in detail in the paper, We have also provided a full description of experiments, to assist others in reproducing our experiments. Additionally, datasets, such as LiveCC, StreamingBench, are publicly available, ensuring consistent and reproducible evaluation results. We believe these measures will enable other researchers to reproduce our work and further advance the field.

### A.3  LLM USAGE

Large Language Models (LLMs) were used to aid in the writing and polishing of the manuscript. Specifically, we used an LLM to assist in refining the language, improving readability, and ensuring clarity in various sections of the paper. The model helped with tasks such as sentence rephrasing,grammar checking, and enhancing the overall flow of the text. It is important to note that the LLM was not involved in the ideation, research methodology, or experimental design. All research concepts, ideas, and analyses were developed and conducted by the authors. The contributions of the LLM were solely focused on improving the linguistic quality of the paper, with no involvement in the scientific content or data analysis. The authors take full responsibility for the content of the manuscript, including any text generated or polished by the LLM. We have ensured that the LLM-generated text adheres to ethical guidelines and does not contribute to plagiarism or scientific misconduct.

# B IMPLEMENTATION DETAILS

## B.1 FAST INFERENCE MECHANISM DETAILS

To ensure low latency in fully online streaming scenarios, we introduce a Fast Inference Mechanism (FIM) designed to reduce computational redundancy. Specifically, we utilize an HSV (Hue, Saturation, Value) color histogram comparison to measure the visual similarity between the current frame $f_t$ and the previous frame $f_{t-1}$. This metric is computationally lightweight, allowing for rapid calculation with negligible overhead. Based on a similarity score $S(f_t, f_{t-1})$ and a pre-defined threshold $\tau$, the mechanism operates in two distinct modes.

**Skip Logic ($S > \tau$).** When the histogram similarity score exceeds $\tau$, the system identifies the current frame as redundant. In such cases, we bypass the Large Language Model's (LLM) forward pass to minimize computational cost, defaulting the output to a `<|silent|>` state. Crucially, to ensure that the model retains a complete understanding of the temporal context, the visual token of the skipped frame is still encoded and pushed into the context window, maintaining temporal continuity without incurring the cost of autoregressive generation.

**Active Inference ($S \leq \tau$).** Conversely, when the similarity falls below $\tau$, it indicates a significant visual change or a potential event trigger. Only in these instances do we perform the full forward pass to calculate the probability distribution between `<|response|>` and `<|silent|>`. This selective processing strategy effectively filters out static or repetitive visual information, significantly reducing the average inference latency.

**Threshold Tuning and Ablation Study.** The similarity threshold $\tau$ acts as a critical hyperparameter controlling the sensitivity of redundancy detection. We determined $\tau$ empirically by evaluating a range of candidate values on our validation set, aiming to identify the optimal trade-off point where inference latency is minimized while the F1-score remains stable. As presented in Table 12, lowering $\tau$ increases throughput but risks missing subtle visual cues. We selected $\tau = 0.98$ as the optimal setting, which improves throughput from 2.36 FPS to 3.08 FPS (+30%) with a negligible drop in Overall F1 (-0.3%).

## B.2 TRIGGER STATE MANAGEMENT (TSM) MODULE

The Trigger State Management (TSM) module is designed to decouple textual interaction history from the high-bandwidth visual stream. Below we elaborate on its motivation and the specific mechanism used to record and analyze queries.

**Motivation: Decoupling Visual and Textual Context.** Standard Multimodal LLMs often struggle in streaming scenarios where continuous video frames and QA pairs are interleaved. The massive influx of visual tokens competes with the textual history, causing the model to lose track of user intent. To address this, we decouple the interaction logic, using a lightweight text-only model to handle dialogue state management explicitly.

**Mechanism: Structured Queue and Relevance Analysis.** As described in the methodology, the TSM operates through a systematic process to ensure consistency across interactions:

- **Structured Queue:** We maintain a structured queue to record queries and their corresponding generated answers. This serves as the external memory for the dialogue history.
- **Relevance Analysis:** Upon the arrival of a new trigger, we leverage a lightweight off-the-shelf model (e.g., Qwen3-1.7B) to analyze its relevance to historical triggers and previously generated answers stored in the queue. This process specifically identifies potential dependencies, contextual links, or contradictions between the current event and past interactions.
- **Query Enrichment:** The derived relational information is then appended to the current query representation. Finally, this enriched query is forwarded to the main model. This step is crucial as it enables the main model to make context-aware predictions that remain consistent across multiple triggers and interactions.

**Table 12:** Effect of FIM threshold $\tau$ on throughput and accuracy. Values in parentheses denote the absolute change relative to the setting without FIM.

| Threshold $\tau$ | Throughput (FPS) | Overall F1 (%) | Recall (%) | Precision (%) |
|---|---|---|---|---|
| w/o FIM | 2.36 | 37.5 | 33.8 | 42.2 |
| $\tau = 0.98$ | 3.08 (+0.72) | 37.2 (-0.3) | 33.0 (-0.8) | 42.8 (+0.6) |
| $\tau = 0.95$ | 3.81 (+1.45) | 36.7 (-0.8) | 31.8 (-2.0) | 43.4 (+1.2) |
| $\tau = 0.92$ | 5.16 (+2.80) | 33.9 (-3.6) | 28.6 (-5.2) | 44.5 (+2.3) |

## B.3 TRIGGER STATE MANAGEMENT (TSM) MODULE DETAILS

The Trigger State Management (TSM) module is designed to decouple textual interaction history from the high-bandwidth visual stream. Below we elaborate on its motivation and the specific event-driven mechanism.

**Motivation: Decoupling Visual and Textual Context.**   Standard Multimodal LLMs often struggle in streaming scenarios where continuous video frames and QA pairs are interleaved. The massive influx of visual tokens competes with the textual history, causing the model to lose track of user intent. To address this, we decouple the interaction logic, using a lightweight text-only model to handle dialogue state management explicitly.

**Event-Driven Mechanism.**   This module implements the Trigger State Management as detailed in our paper. Specifically, we maintain a chronologically ordered structured queue that records the sequence of User Query and Model Response pairs. Upon a new trigger, the model analyzes its relevance to this history to identify dependencies, contextual links, or contradictions. The derived relational information is used to rewrite and enrich the current query for the main backbone.

---

**Illustrative Example.** Consider a sports scenario where the interaction unfolds over a continuous stream:

- **History** ($t_1$): *Model Answer: "The blue player scored a goal."*
- **Video Stream** ($t_1 \rightarrow t_2$): *[Continuous frames inflow]*
- **New Trigger** ($t_2$): *User Query: "What did this player do?"*

Despite the visual context evolving, the TSM module analyzes the history to resolve the dependency and outputs the enriched query:

*Enriched Query: "What did the blue player do?"*

This enables the main model to make context-aware predictions that remain consistent across interactions.

---

## B.4 TRAINING STAGE 1: STREAMING CAPTION PRE-TRAINING

The goal of the first stage is to adapt the model to the streaming video input format and to teach it when to speak and when to remain silent. In this process, the model learns special control schema including `<|response|>`, `<|silent|>`, and `<|end_of_stream|>`. For the model to learn these special placeholders, we designed two tasks.

**Deferred Summarization.** At the beginning of the video, the instruction is given: *"Please summarize the video after watching it (i.e., once the `<|end_of_stream|>` token is received)."*. The model is trained to continuously output `<|silent|>` throughout the entire stream. The `<|response|>`, along with a final summary, is generated at the end.

**Immediate Summarization.** Just before the `<|end_of_stream|>` token, an instruction is inserted: *"Please immediately summarize everything you have seen so far."*. The model is trained to generate an instant recap at that point by switching from `<|silent|>` to `<|response|>`.

## B.5 TRAINING STAGE 2: DIVERSE ONLINE QA FINE-TUNING

The second stage, supervised fine-tuning, teaches the model to handle complex, goal-driven online QA tasks across diverse scenarios. To achieve this, we construct an online QA dataset that integrates multiple task types, enabling the model to learn how to interpret queries and incorporate multimodal evidence.

**Training Sequences.** Each training example is formatted as a concatenated sequence of the query and multimodal context:

$$\texttt{Query} + \langle \texttt{image}_1 \rangle + \langle \texttt{silent} \rangle + \dots + \langle \texttt{image}_n \rangle + \langle \texttt{response} \rangle + \texttt{Answer}.$$

The model is trained with loss applied only on the control tokens `<|silent|>`, `<|response|>`, and the `Answer`, ensuring it learns both when to remain silent and when to generate a response.

**ID Tagging Mechanism.** To support multi-turn QA, we introduce explicit ID tags, such as `<question-id>Q1</question-id>` and `<answer-id>A1</answer-id>`.

Similar to Stage 1, the model is also trained to produce answers after encountering the `<|end_of_stream|>` placeholder. However, unlike Stage 1, where the query is fixed at the beginning, in this stage the query may be inserted at arbitrary positions in the sequence, exposing the model to a richer scenario.

## B.6 TRAINING STAGE 3: REINFORCEMENT LEARNING

Reinforcement Learning (RL) for alignment is applied to overcome the rigidity of the stage 2 training. RL allows the model to learn a robust policy that tolerates small, acceptable variations in response time. Additionally, RL prevents shortcut behavior: for example, emitting long runs of `<|silent|>` that minimize training loss but degrade output quality.

Early alignment efforts in the Reinforcement Learning from Human Feedback (RLHF) pipeline, beginning with Ouyang et al. (2022), employed the Proximal Policy Optimization (PPO) algorithm to train InstructGPT. Subsequent methods have focused on reducing the reliance on costly human annotation. Reinforcement Learning from AI Feedback (RLAIF) Lee et al. (2024) substitutes human raters with a powerful teacher model that generates the necessary preference labels. Direct Preference Optimization (DPO) Rafailov et al. (2023) further streamlines the process by removing the explicit reward model. More recently, approaches have sought to simplify the RL objective. Group Relative Policy Optimization (GRPO) Shao et al. (2024), introduced by DeepSeek, eliminates the value function. GRPO operates by generating multiple candidate responses per prompt, scoring them directly using a reward model, and then applying normalization within this candidate group to calculate relative advantages for the policy updates. Other variants, such as DAPO Yu et al. (2025), address the challenge of stalled learning that can occur when multiple candidate responses receive identical scores. DAPO mitigates this by employing dynamic sampling to diversify candidates and by incorporating token-level gradients derived from normalization across tokens within all responses. Recent advancements have further extended these alignment paradigms to the multimodal domain. Several studies have demonstrated that applying RLHF to Multimodal Large Language Models (MLLMs) significantly enhances their fine-grained visual perception and complex reasoning capabilities (Leng et al., 2025; Huang et al., 2025a; Shen et al., 2025; Feng et al., 2025; Peng et al., 2025; Li et al., 2025b).

We adopt GRPO for two reasons: 1) our reward function is complex and generated outputs are complex, which ensures that multiple candidates rarely receive identical scores, mitigating common learning stalls. 2) We operate in a streaming scenario where overly long generations are undesirable. The GRPO's averaging strategy naturally diminishes the reward as responses become longer, which implicitly discourages verbosity.

### B.6.1 ROLLOUT, MASK, AND LOSS

**Rollout.** At each absolute time step $t = 1, \dots, T$, the model is conditioned on the instruction and the concatenated history up to $t - 1$ (which includes previously generated tokens and externally provided instructions), and autoregressively produces the next token

$$o_t^{\text{pred}} \sim \pi_{\theta_{\text{old}}}\left( \cdot \mid q, \, o_{<t}^{\text{pred}} \right).$$

Repeating this yields a complete multi-turn streaming trajectory

$$\tau = \{o_1^{\mathrm{pred}}, \ldots, o_T^{\mathrm{pred}}\}.$$

For GRPO, we sample a group of $G$ trajectories $\{\tau_i\}_{i=1}^G$ under $\pi_{\theta_{\mathrm{old}}}$ for the same instruction $q$.

**Masking.** We optimize on *model-generated tokens* only, i.e., tokens in the vocabulary $\mathcal{V}$. Let the binary mask $m_t \in \{0, 1\}$ be

$$m_t = \mathbf{1}\left[o_t^{\mathrm{pred}} \in \mathcal{V}\right],$$

so that both `<|silent|>` and `<|response|><answer-id>` **A** `</answer-id>` are included ($m_t = 1$), while externally provided instruction/state tokens (the concatenated stage at each round) are excluded ($m_t = 0$).

**Loss.** For each group $i$ and time step $t$, define the importance ratio

$$r_{i,t}(\theta) = \frac{\pi_\theta\left(o_{i,t}^{\mathrm{pred}} \mid q, o_{i,<t}^{\mathrm{pred}}\right)}{\pi_{\theta_{\mathrm{old}}}\left(o_{i,t}^{\mathrm{pred}} \mid q, o_{i,<t}^{\mathrm{pred}}\right)}.$$

Let $\hat{A}_{i,t}$ denote the group-relative advantage derived from the per-time-step reward construction in the previous subsection. The GRPO objective applied with masking is $J_{\mathrm{GRPO}}(\theta) = \mathbb{E}_{q \sim P(Q), \{\tau_i\}_{i=1}^G \sim \pi_{\theta_{\mathrm{old}}}}$

$$\left[\frac{1}{G}\sum_{i=1}^G \frac{1}{|o_i|}\sum_{t=1}^T m_{i,t}\min\left(r_{i,t}(\theta)\,\hat{A}_{i,t}, \mathrm{clip}\left(r_{i,t}(\theta), 1-\varepsilon, 1+\varepsilon\right)\hat{A}_{i,t}\right)\right] - \beta\,D_{\mathrm{KL}}\left(\pi_\theta \,\|\, \pi_{\mathrm{ref}}\right).$$

The mask $m_{i,t}$ ensures that gradients flow through all model-generated tokens, while the concatenated instruction tokens do not contribute to the loss.

## C  DATASET CONSTRUCTION AND DETAILS

### C.1  DATA DETAILS

To construct our dataset, we designed a systematic, multi-stage pipeline to synthesize QA pairs for all three task categories from diverse, timestamped sources. This process ensures our final data covers a range of trigger granularities and task complexities. The specific synthesis strategy for each task category is detailed below.

**Synthesis Pipeline for Temporal Grounded Question Answering (TGQA)** For the synthesis of our event-grounding tasks, we employ a multi-source strategy that leverages datasets with segment-level, frame-level, and spatio-temporal annotations.

- **From Densely Segmented Datasets (e.g., *VRiPT, Shot2Story*):** For these datasets, we employ a Large Language Model (LLM) to analyze the sequence of annotated events $\{E_1, E_2, \ldots, E_n\}$. The LLM's task is to first autonomously select a salient event $E_k$ to serve as the trigger. It then synthesizes a QA pair where the question is grounded in the historical context provided by the preceding annotations $\{E_1, E_2, \ldots, E_{k-1}\}$. This method is powerful enough to create a wide range of complex questions, including those requiring *visual perception*, *information recall*, and *reasoning*.

- **From Spatio-temporal Localization Datasets (e.g., *DiDeMo, Charades*):** For these datasets, which provide a natural language query but lack dense contextual annotations, our approach is to have an LLM rephrase the existing query into a forward-looking, trigger-based format. The trigger point becomes the event described in the original query. Due to the limited context, the synthesized QA pairs are primarily *simple triggers*, such as state confirmations or instruction-following tasks (e.g., responding with a specific phrase upon observing an event).

- **From Densely Frame-level Datasets (e.g., *Ego4D*):** For these datasets, we leverage the discrete, fine-grained nature of the data. An LLM is prompted to synthesize *simple event-trigger QA pairs* directly from these individual frame-level descriptions.

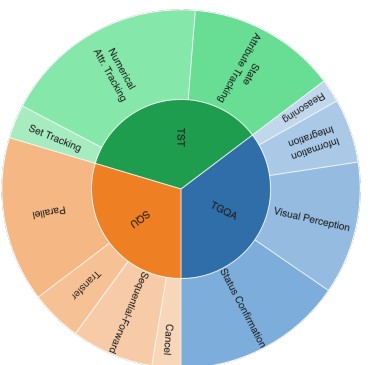

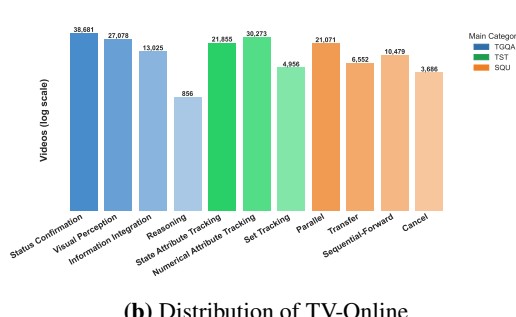

(b) Distribution of TV-Online

(a) Catergories of TV-Online

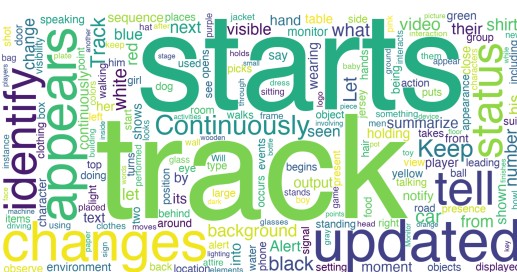

**Figure 7:** Word Cloud of TV-Online

**Synthesis Pipeline for Temporal State Tracking (TST) QA**   For the synthesis of our state-tracking tasks, we utilize densely annotated datasets at both the segment and frame levels.

- **Data Filtering:** We first apply a filtering process to the source videos, excluding those that are too short or have sparse and uninformative annotations. This step ensures that the remaining data is sufficiently rich to support the generation of multi-answer QA formats, which are a core component of this task category.

- **Attribute-based Tracking Synthesis:** For single-entity tracking, we employ a two-stage, LLM-driven process. First, the LLM analyzes the full annotation sequence to extract candidate objects for tracking ($O$), which can include entities, events, processes, or attributes. Subsequently, based on a selected object $O$, the LLM generates a continuous tracking question. Finally, the LLM scans the entire annotation timeline again to identify all segments related to $O$ and synthesizes the corresponding answer stream.

**Synthesis Pipeline for Sequential Query Understanding (SQU) QA**   To construct our multi-turn QA data, we designed several strategies to simulate realistic, context-dependent dialogues.

- **Independent (Parallel) Queries:** For scenarios where queries are unrelated, we create dialogues by randomly selecting 2-3 existing QA pairs from our previously generated TGQA or TST data. The first question (Q1) is anchored to the beginning of the video, while subsequent questions (Q2, Q3) are inserted at later, randomized timestamps.

- **Query-Dependent (Instruction Modification):** For scenarios where a later query modifies a previous one, we use an LLM to generate the subsequent turns. Given an initial question (Q1), the LLM is prompted to generate a second question (Q2) that logically alters Q1, following one of several modification patterns: specialization (refinement), generalization, deletion, or correction.

- **Answer-Dependent (Follow-up Queries):** To simulate dialogues where a user's question is a direct follow-up to a model's answer, we generate Q2 based on the content of A1.
  - *Retrospective follow-ups* focus on content preceding A1's timestamp and require an immediate response.
  - *Prospective (forward-looking) follow-ups* are also dependent on A1 but require a delayed response based on subsequent annotations.

  To ensure data quality, this process is conditional: the LLM is instructed not to generate a follow-up if the future video content does not logically support one. Furthermore, to evaluate the model's coreference resolution capabilities, we instruct the LLM to establish a clear "anchor" entity from a prior answer, allowing subsequent queries to refer to it using pronouns.

The distributions of TV-Online can be seen in Table 6a and Table 6b. The word cloud of our dataset can be seen in Figure 7.

## C.2 DOMAIN DISTRIBUTION AND DIVERSITY

To ensure semantic diversity and prevent coverage bias, we curated videos from a wide range of domains. As shown in Table 13, TV-Online categorizes content into five macro-domains, covering everything from daily household activities to professional sports. This diversity ensures that the benchmark meaningfully tests generalizable online video understanding rather than surface patterns specific to a single domain.

**Table 13:** Macro-domain and category distribution in TV-Online.

| Macro-Domain | Detailed Categories Covered | Primary Data Sources |
|---|---|---|
| **1. Daily Life & Household** | Baking, Laundry, Cooking, Cleaning, Knitting, Indoor Object Interaction | Charades, Ego4D, Vript |
| **2. Social, Travel & Lifestyle** | Vlogs, Travel, Pets, Autos, Casual Events, Family Moments | DiDeMo, Shot2Story, Vript |
| **3. Arts, Entertainment & Gaming** | Film, Animation, Comedy, Gaming, Reading | Shot2Story, Vript |
| **4. Knowledge, Science & News** | Science, Technology, Education, Politics | Shot2Story, Vript |
| **5. Sports & Action** | Professional Sports, Casual Athletics, High-motion Activities | All datasets |

## C.3 HUMAN AUDIT AND QUALITY ASSURANCE

While our pipeline is automated, we conducted a rigorous human audit to validate the quality of the generated data. We randomly sampled 500 QA pairs from the test set, which were evaluated by three independent annotators—graduate students with prior experience in video understanding and multimodal learning tasks. All annotators share a similar academic background in computer science

**Table 14:** Statistics of the TV-Online evaluation set.

| Task | Videos | Questions | Answers |
|------|--------|-----------|---------|
| Basic | 1150 | 1150 | 1150 |
| Intermediate | 900 | 900 | 3073 |
| Advanced | 400 | 821 | 1255 |
| Expert | 1000 | 2000 | 1600 |
| **Total** | **3450** | **4871** | **7078** |

and AI-related fields and underwent the same training protocol for judging semantic grounding and temporal alignment, ensuring a consistent interpretation of the evaluation rubric and minimizing variance due to cultural or subjective differences.

Each sample was assessed based on three criteria: (1) clarity of the question, (2) relevance to the video content, and (3) accuracy of the timestamped answer. Overall, 94.2% of the sampled QA pairs were accepted by at least two annotators, indicating that the vast majority exhibit both correct semantic grounding and accurate temporal alignment. The audit further yielded a Fleiss' Kappa score of 0.86, reflecting strong inter-annotator agreement despite independent review. This high consistency suggests that the QA pairs are not only high-quality but also unambiguous and easy to interpret. Taken together, the high acceptance rate and strong agreement provide compelling evidence that our automated multi-stage pipeline produces reliable annotations with minimal noise or ambiguity propagated into the benchmark.

### C.4 TEST SET STATISTICS

To ensure full transparency regarding our experimental setup, we provide a detailed breakdown of the TV-Online evaluation set. We strictly enforce a rigorous training-testing split: the final split guarantees no overlap between the training and test sets. Crucially, all training videos are sourced exclusively from the training splits of the original source datasets, ensuring that no testing videos from external benchmarks are included in our training data. This prevents any potential data leakage and ensures a fair evaluation.

The detailed statistics of the TV-Online evaluation set are presented in Table 14. The benchmark is categorized into four difficulty levels—Basic, Intermediate, Advanced, and Expert—to comprehensively evaluate model capabilities ranging from simple temporal grounding to complex, multi-trigger reasoning.

### C.5 TRAINING DATA SCALE ANALYSIS

To provide transparency regarding our progressive training pipeline and to investigate how the scale of training data influences model performance, we detail the data composition for each stage and conduct an ablation study on training steps.

**Training Data Composition.** Our training pipeline consists of three progressive stages: Pre-training (large-scale alignment), Supervised Fine-Tuning (SFT), and Reinforcement Learning (RL). Table 15 details the exact composition of videos and textual instructions used in each stage. The data selection shifts from broad, open-domain sources in Stage 1 to the domain-specific, high-quality TV-Online dataset in Stages 2 and 3.

**Impact of Training Steps (Scaling Analysis).** To demonstrate the benefit of data scaling, we analyzed the model's performance at different training steps during the SFT stage. As shown in Table 16, the model exhibits consistent performance gains across all tasks (TGQA, TST, SQU) as training progresses and more data samples are consumed. This trend confirms that our framework effectively benefits from the scale of the dataset, with the final model achieving the best trade-off between tasks.

**Table 15:** Statistics of training data across three stages.

| Stage | Data Sources (Videos) | Text Statistics |
|---|---|---|
| **Stage 1: Pre-training** | • Shot2Story: 27.8k
• Vript: 30.7k | • Instructions: 58k
• Answers: 338k |
| **Stage 2: SFT** | • TV-Online (Train): 48k | • Instructions: 256k
• Answers: 500k |
| **Stage 3: RL** | • TV-Online (Subset): 1k[*] | • Instructions: 1.8k
• Answers: 3.0k |

**Table 16:** Ablation study on the impact of training data scale.

| Training Steps | TGQA | TST | SQU |
|---|---|---|---|
| **3000 steps** | 40.1 | 24.1 | 24.8 |
| **5000 steps** | 43.6 | 28.3 | 28.5 |
| **Final** | 44.5 | 30.0 | 29.8 |

## C.6 DATA EXAMPLES

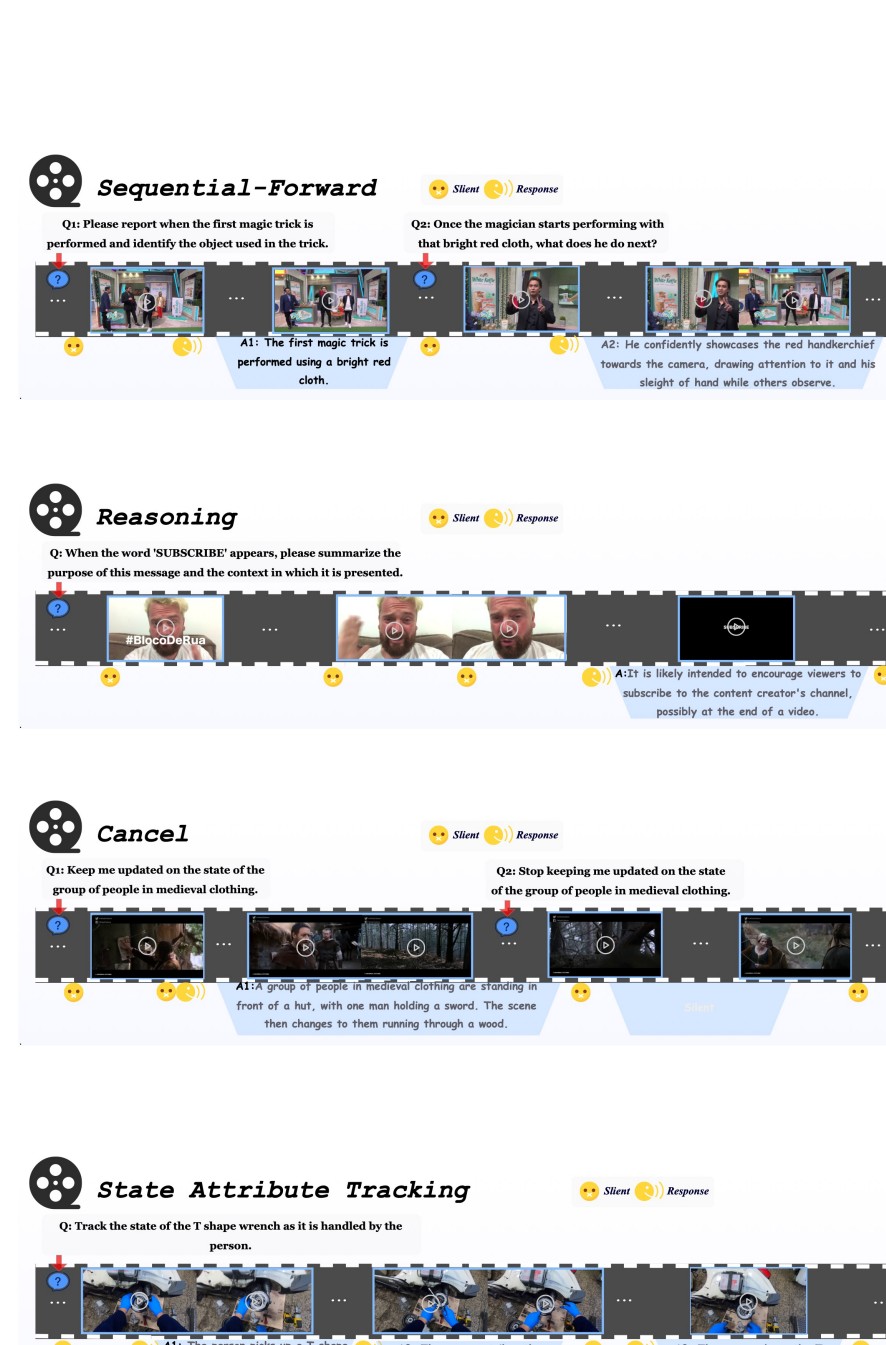

**Figure 8:** Some QA Examples of TV-Online

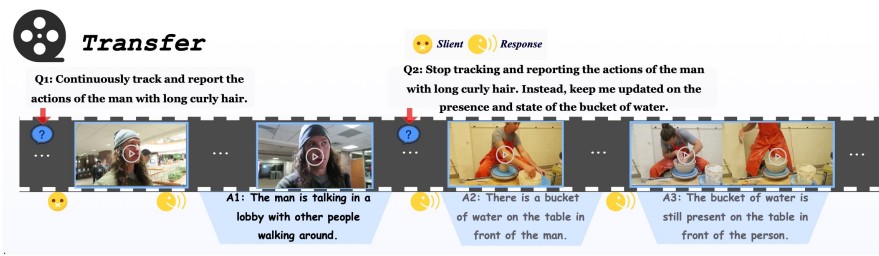

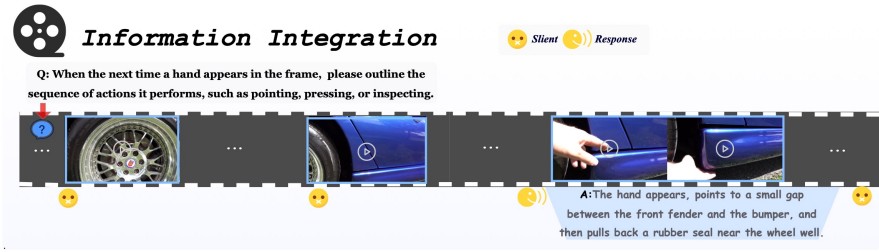

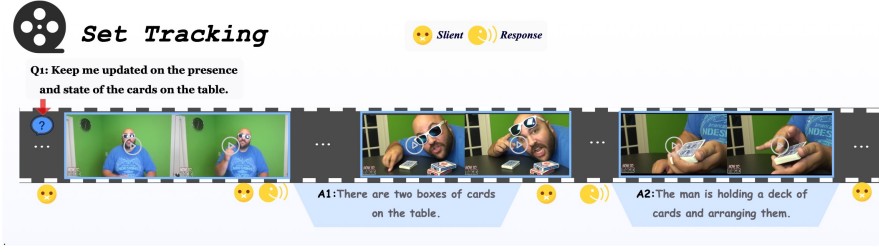

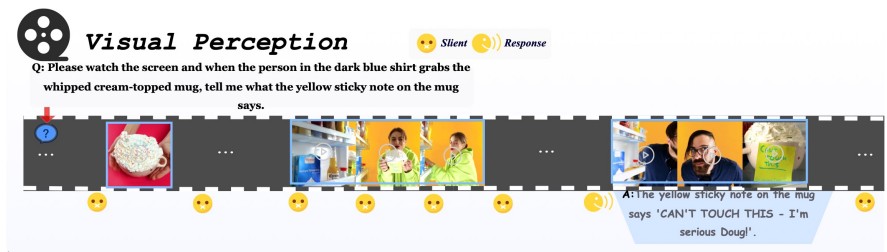

**Figure 9:** Some QA Examples of TV-Online

# D    ADDITIONAL EXPERIMENTS

## D.1    GENERALIZATION ON OTHER BENCHMARKS

To demonstrate the strong generalization capability of our framework beyond the specific domain of TV-Online, we extended our evaluation to a comprehensive suite of streaming video understanding benchmarks, including **ViSpeak-Bench**, **ProactiveVideoQA**, **StreamingBench**, **SVBench**, **OVO-Bench**, and **OmniMMI**. As detailed below, TOM consistently achieves top-tier performance on proactive response tasks and various online multi-turn dialogue scenarios across these benchmarks.

To ensure the rigor and fairness of our comparative analysis, we adopted the following protocols: **Experimental Setup for Fair Comparison.** We strictly adhered to the official evaluation protocols for all benchmarks (with a necessary adaptation for ProactiveVideoQA, detailed below). Furthermore, to ensure a fair comparison with larger state-of-the-art baselines, we trained a 7B-parameter variant (**ToM-7B**) under the same settings as our main model. For detailed training configurations and a comprehensive performance comparison between the 3B and 7B models, please refer to Section D.3.

**ViSpeak-Bench.**    On ViSpeak-Bench, we achieved the second-best performance, surpassed only by the audio-enabled ViSpeak (s3), while significantly outperforming our base model, Qwen2.5-VL-7B. In terms of Time Accuracy, we observed a notable gap between our ToM model and ViSpeak (s3), particularly in the Anomaly Warning (AW) and Humor Reaction (HR) tasks. We attribute this performance gap to the inherent ambiguity of these "non-grounding" tasks. Unlike clear physical events, concepts like "anomaly" and "humor" lack objective onset thresholds. Consequently, our model adopts a **conservative inference strategy**—waiting for sufficient visual context (e.g., an event fully unfolding) to ensure reliability. While this impacts strict "Time Accuracy," it effectively avoids premature hallucinations. Crucially, this difficulty is intrinsic to the tasks rather than specific to our model; as reported in the ViSpeak paper Fu et al. (2025b), human participants also achieved their lowest scores on AW and HR, struggling with the same subjectivity. In contrast, our superior performance on **Visual Interruption (VI)** demonstrates the model's robustness in handling complex, intent-driven interactions where clear communicative signals exist.

**ProactiveVideoQA.**    To further demonstrate the generalization capability and superiority of our proposed TOM in proactive settings, we conducted additional evaluations on the ProactiveVideoQA benchmark (Wang et al., 2025b). Since the official benchmark lacks a standardized setting for streaming inference, we established a **strict online evaluation protocol**: (1) **Frame-by-Frame Inference:** The model processes the video sequentially frame-by-frame; (2) **Autonomous Decision Making:** At each frame, TOM autonomously outputs either a special `<|silent|>` token (to remain silent) or a `<|response|>` token followed by the answer; and (3) **Metric Consistency:** We collect all responses and their timestamps upon video completion, utilizing the official evaluation scripts with GPT-4.1 as the judge to ensure consistency with the reported results in the original paper. We compared TOM-3B and TOM-7B against strong baselines including VideoLLM-Online and MMDuet. As shown in Table 9, TOM-7B achieves state-of-the-art overall performance (**37.35**), outperforming MMDuet (34.68) and VideoLLM-Online (23.7). Notably, our model shows substantial advantages in open-domain web videos (WEB) and TV shows (TV), validating the effectiveness and generalizability of our proposed model and data strategy across diverse online scenarios.

**StreamingBench and OVO-Bench.**    As detailed in Table 10, ToM-7B demonstrates superior generalization capabilities. On StreamingBench, it achieves a state-of-the-art Overall Score of **48.78**, significantly outperforming strong baselines such as DispiderQian et al. (2025) (45.63) and Qwen2.5-VL (46.08). Notably, the model leads in *Contextual Understanding* (34.20), which provides empirical validation for the TSM module's ability to maintain robust long-term context over extended streams. Furthermore, on the OVO-Bench *Forward Active Responding* task, ToM surpasses all baselines with a score of **54.68%**. This superiority confirms that our Event-Driven mechanism effectively facilitates precise, real-time proactive interaction, ensuring the model's adaptability to diverse and open-ended streaming scenarios.

**Table 17:** Results on OmniMMI.

| Method | Params | Frames | SG 1st | SG 2nd | SG 3rd | SG avg | AP | MD 1st | MD 2nd | MD 3rd | MD avg |
|---|---|---|---|---|---|---|---|---|---|---|---|
| **VideoLLM-online**[*] (Chen et al., 2024) | 8B | 1 fps | 18.00 | 4.75 | 0.00 | 4.67 | 35.00 | 18.00 | 4.75 | 0.00 | 1.33 |
| **IXC2.5-OL**[*] (Zhang et al., 2024b) | 7B | 512 | 40.33 | 5.08 | 0.00 | 4.03 | 30.50 | 26.00 | 4.50 | 1.52 | 4.00 |
| **Qwen2.5-VL** (Bai et al., 2025) | 7B | 32 / 1 fps | 38.67 | 4.40 | 1.00 | 4.30 | 28.50 | 27.00 | 7.90 | 3.40 | 7.00 |
| **M4**[*] (Wang et al., 2025c) | 7B | 32 / 1 fps | 35.67 | 6.44 | 1.87 | 5.67 | 33.50 | 35.67 | 6.44 | 1.87 | 1.67 |
| **ToM (ours)** | 7B | 32 / 1 fps | **41.00** | **7.80** | 1.60 | **6.30** | **35.00** | 30.60 | **8.60** | **3.50** | **8.00** |

**Table 18:** Results on SVBench.

| Model Name | Semantic Accuracy | Contextual Coherence | Logical Consistency | Temporal Understanding | Informational Completeness | Overall |
|---|---|---|---|---|---|---|
| **GPT4o**[*] (OpenAI et al., 2024) | 5.811 | 6.476 | 7.067 | 6.751 | 5.586 | 6.260 |
| **LiveStar**[*] (Yang et al., 2025d) | 5.258 | 5.984 | 6.545 | 6.459 | 5.135 | 5.759 |
| **MiniCPM-V-2.6**[*] (Yao et al., 2024) | 5.163 | 5.949 | 6.542 | 6.172 | 4.993 | 5.669 |
| **ToM-7B (ours)** | 5.054 | 5.730 | 6.424 | 5.874 | 5.035 | 5.503 |
| **Gemini-1.5-Pro**[*] (Team et al., 2024) | 4.908 | 5.621 | 6.215 | 5.848 | 4.766 | 5.363 |
| **Qwen2.5-VL-7B** (Bai et al., 2025) | 5.001 | 5.552 | 6.228 | 5.264 | 4.788 | 5.287 |
| **InternLM-XComposer2.5**[*] (Zhang et al., 2024b) | 4.307 | 5.038 | 5.933 | 5.452 | 4.234 | 4.856 |
| **video_llama2**[*] (Cheng et al., 2024) | 4.251 | 4.995 | 5.596 | 5.218 | 4.153 | 4.721 |
| **VILA**[*] (Lin et al., 2024a) | 4.311 | 4.927 | 5.563 | 5.253 | 4.139 | 4.707 |
| **streamingcaption**[*] (Zhou et al., 2024) | 3.753 | 4.488 | 5.434 | 5.003 | 3.642 | 4.272 |
| **Flash-VStream**[*] (Zhang et al., 2024a) | 3.766 | 4.482 | 5.103 | 4.810 | 3.791 | 4.271 |

**OmniMMI.** To further assess the generalization of our trigger-centric online model beyond TV-Online, we evaluate ToM on the streaming understanding subsets of the OmniMMI benchmark, including *Dynamic State Grounding* (SG), *Action Planning* (AP), and *Multi-turn Dependency Reasoning* (MD). These tasks are explicitly designed to probe online multi-turn capabilities: the model must continuously update its understanding of the evolving video state (SG) and maintain cross-turn consistency when later questions depend on earlier answers (MD).

As summarized in Table 17, ToM (7B) consistently outperforms the strong streaming baseline M4 (7B) under the same frame budget (32 / 1 fps). On SG, ToM achieves a higher average score (SG avg: 6.30 vs. 5.67), indicating more reliable tracking of dynamic visual states over time. On AP, ToM also improves over M4 (35.0 vs. 33.5), and the gap is most pronounced on MD: ToM attains a substantially better multi-turn dependency score (MD avg: 8.00 vs. 1.67). Although the absolute numbers on OmniMMI remain modest—highlighting the inherent difficulty of long-horizon streaming interaction—these consistent gains over M4 on all online multi-turn subtasks demonstrate that our trigger-aware state management and protocol-level design generalize well to a distinct benchmark with different data distribution and interaction patterns.

**SVBench.** SVBench Yang et al. (2025c) is a pioneering benchmark designed to evaluate LVLMs in long-context streaming video understanding. Unlike traditional benchmarks that focus on isolated queries, SVBench features temporal multi-turn question-answering chains derived from 1,353 streaming videos, specifically assessing the capacity to sustain temporal reasoning and maintain dialogue coherence over extended durations. To ensure rigorous evaluation, we strictly adhered to the official **Dialogue Evaluation** protocol and submitted our model predictions to the official leaderboard for verification. The returned results, as presented in Table 18, demonstrate the superior capability of our model. Notably, ToM-7B achieves an Overall Score of **5.503**, significantly outperforming strong open-source baselines such as Qwen2.5-VL-7B (5.287) and VideoLLaMA2 (4.721). More importantly, our 7B model surpasses the closed-source commercial model **Gemini-1.5-Pro** (5.363), highlighting its exceptional ability to handle complex context maintenance and logical consistency in continuous multi-turn dialogues.

### D.2 ABLATION STUDY ON FPS AND TOKEN BUDGET

To investigate the trade-off between inference efficiency and model performance, we conducted ablation studies varying the frame sampling rate (FPS) and the visual token budget. The results are summarized in Table 19.

**Table 19:** Effect of FPS and visual token budget on precision, recall, and F1.

| FPS / Visual Tokens Setting | Precision | Recall | Overall F1 |
|---|---|---|---|
| **0.5 / 48** | 43.1 (+1.1) | 31.2 (-2.6) | 36.2 (-1.3) |
| **1.0 / 48 (Default)** | 42.0 | 33.8 | 37.5 |
| **2.0 / 48** | 40.3 (-1.7) | 35.5 (+1.7) | 37.7 (+0.2) |
| **1.0 / 24** | 40.7 (-1.3) | 32.5 (-1.3) | 36.1 (-1.4) |

**Table 20:** Comparison of ToM models with different parameter scales on TV-Online.

| Model | TGQA | TST | SQU | Overall |
|---|---|---|---|---|
| **ToM-3B** | 45.2 | 35.0 | 31.8 | 37.5 |
| **ToM-7B** | 50.8 | 37.5 | 36.7 | 41.6 |

We evaluated the model at 0.5, 1.0, and 2.0 FPS. As observed, the model demonstrates remarkable robustness to varying sampling rates. Specifically, increasing the FPS leads to a slight increase in Recall (due to observing more frames within the fixed memory window $W$) but a minor decrease in Precision (due to more active responses). Consequently, the Overall F1 score remains stable. This stability is largely attributed to the Fast Inference Mechanism (FIM), which effectively detects and skips the increased volume of redundant frames at higher FPS, ensuring consistent model behavior without overwhelming the inference pipeline.

**Feasibility under Tight Token Budgets.** To assess the model's adaptability to resource-constrained environments, we reduced the visual token budget from 48 to 24 per frame without retraining. The results show only a marginal performance decrease (Overall F1 drops from 37.5 to 36.1). This robustness suggests that the trigger-centric task prioritizes high-level temporal logic over fine-grained spatial details. It also validates the strong generalization capability of our backbone (Qwen2.5-VL) to interpret compressed visual representations, highlighting the potential for adaptive budgets in on-device deployment.

### D.3 ABLATION STUDY ON MODEL SIZE (3B VS 7B)

To verify the scalability of our proposed framework and ensure a fair comparison with larger state-of-the-art baselines, we trained a 7B-parameter variant (**ToM-7B**).

**Training Implementation Details.** Unlike the full fine-tuning used for ToM-3B, we employed Low-Rank Adaptation (LoRA)Hu et al. (2022) for the 7B model to mitigate potential catastrophic forgetting and maintain training efficiency. Specifically, we unfroze the `lm_head` and `embed_tokens` layers to ensure proper alignment of the vocabulary space, while applying LoRA adapters to the attention layers. All other training configurations—including the three-stage curriculum, learning rates, and data mixtures—were kept strictly identical to the ToM-3B setup.

**Performance Scalability.** As presented in Table 20, scaling the model size yields consistent performance improvements across all task categories. ToM-7B outperforms the 3B variant with substantial margins, boosting the Overall F1 score from 37.5 to 41.6. Notably, the improvement in TGQA (+5.6) and SQU (+4.9) highlights that larger model capacity effectively enhances both temporal grounding precision and multi-turn reasoning capabilities. These results confirm that our data construction pipeline and training framework are scalable and can effectively leverage stronger backbones.

# E  DISCUSSION AND LIMITATIONS

## E.1  COMPARISON WITH OFFLINE MODELS

We acknowledge that there is a performance gap between our streaming model (ToM) and offline baselines, particularly in the TGQA task. However, we attribute this primarily to a **Protocol-Induced Gap** stemming from fundamentally different evaluation settings.

**Distinct Evaluation Protocols.** The discrepancy in performance is largely attributable to the disparate information available to the models under the two protocols:

- **Offline Setting:** Offline models typically operate under a "pre-segmented" protocol. They are fed video clips that are cropped precisely to the ground-truth timestamps of the event (or have access to the full video with future context). Consequently, the offline model's task is simplified to semantic reasoning within a perfect temporal window, effectively bypassing the challenge of temporal localization.

- **Online Setting:** In contrast, ToM operates in a strictly online, frame-by-frame streaming scenario. It has **no access** to future frames or ground-truth start/end times. The model must autonomously determine both *when* an event occurs (trigger detection) and *what* it entails (content generation) solely based on historical context.

**Contextual Complexity in Online Inference.** Beyond temporal localization, the online setting imposes significantly higher cognitive demands, especially for the TGQA task. While offline models generally process "clean" inputs (a curated video segment paired with a specific query), online models face a complex, **interleaved context** where continuous video frames, historical user queries, and previous model responses are mixed in the stream. This forces the model to actively filter out irrelevant history and manage the interference between interleaved visual and textual modalities to accurately identify the correct trigger. This dynamic and noisy context makes the online task inherently more challenging than the pure semantic reasoning required in the offline setting. Given these differences, the performance of offline models effectively serves as a **theoretical upper bound for semantic reasoning**. The observed gap in TGQA scores highlights the immense difficulty introduced by the unconstrained streaming setting—where errors in timing and the burden of managing complex, interleaved contexts directly penalize the score—rather than a deficiency in the model's representation capabilities.

## E.2  LIMITATIONS

While our framework establishes a strong baseline for trigger-centric online video understanding, we identify distinct limitations regarding the inference mechanism and modality scope.

**Failure Modes of Fast Inference.** We acknowledge that the Fast Inference Mechanism (FIM) may miss triggers that occur during visually static but semantically important moments. This is a natural limitation of using HSV-based visual similarity for redundancy detection: when frames are nearly identical, FIM intentionally skips them to minimize latency. Consequently, if a scene remains visually unchanged while a critical event unfolds (e.g., a subtle dialogue shift without significant motion), the system may bypass the forward pass, potentially leading to missed triggers.

**Absence of Audio Modality.** The limitation of FIM is often exacerbated by the vision-only nature of our current setup. Real-world events are frequently driven by non-visual cues—such as speech, acoustic reactions, or environmental sounds—which naturally occur during visually static intervals. Indeed, our evaluation on the ViSpeak benchmark supports this observation: performance gaps in categories like Anomaly Warning (AW) and Humor Reaction (HR) are largely attributable to the model's inability to perceive acoustic signals (e.g., crashes or laughter) (Fu et al., 2025b). We view multimodal expansion—specifically the integration of synchronized audio streams into the protocol-tagging scheme—as a promising direction to mitigate these limitations and achieve greater robustness in complex, multisensory streaming scenarios.

# F  FUTURE DIRECTIONS

Building upon the foundational capabilities of TOM, we envision three strategic directions to address current limitations and further advance trigger-centric online video understanding.

**Multimodal Trigger Grounding.**  To transcend vision-only limitations, we plan to extend the protocol-tagging scheme to the auditory domain. By introducing audio-specific control tokens (e.g., `<|audio_trigger|>`), the model can learn to ground events in both visual changes and acoustic signals (e.g., detecting crashes or interruptions even in static scenes). This extension directly addresses performance gaps in audio-dependent scenarios, as observed in benchmarks like ViSpeak Fu et al. (2025b), enabling truly holistic streaming perception.

**Adaptive Inference Policies.**  Moving beyond the static HSV threshold in FIM, future work will explore **learnable skipping policies**. We aim to train a lightweight policy network (or leverage internal LLM confidence scores) to dynamically adjust frame sampling rates and token budgets based on semantic entropy. This approach seeks to optimize the latency-accuracy Pareto frontier, allowing the system to bypass non-informative segments while concentrating computational resources on complex, high-entropy moments.