# OpenReview forum: "Learning to Respond:  A Large-Scale Benchmark and Progressive Learning Framework for Trigger-Centric Online Video Understanding"
_ICLR.cc/2026/Conference — Submitted to ICLR 2026_

### Official Review · Reviewer_67ms · 2025-10-25

**Soundness:** 2
**Presentation:** 2
**Contribution:** 2
**Rating:** 4
**Confidence:** 5

**Summary:**

The paper introduces TV-Online, a trigger-centric benchmark for online video understanding with ~50K videos, ~200K questions, and ~500K time-stamped answers organized into hierarchical tasks—TGQA, TST, and SQU. It proposes TOM, an agent-style streaming model with protocol-level tags, fast inference, and queue-based state management, trained via a progressive pipeline (pretrain → fine-tune → RL). A unified metric holistically scores semantic correctness, coverage completeness, and temporal latency. Experiments and ablations on TV-Online report strong overall F1 and sizable gains from ID tagging and state management.

**Strengths:**

**1. Trigger-centric dataset & metric.** A large, hierarchical benchmark with explicit temporal triggers and a three-axis metric (semantic, coverage, temporal) targets online responsiveness beyond offline QA.

**2. Agent-style TOM with protocol & memory.** Special tags (<|response|>, <|silent|>, IDs), fast inference, and Trigger State Management align frame-by-frame decisions and multi-trigger coherence.

**3. Progressive training with RL.** A staged pipeline plus rewards (coverage/precision, tag reward, hallucination penalty) improves Overall/SQU F1; ID tags alone add +11.8 SQU and +6.8 Overall.

**Weaknesses:**

**1. Novelty of method.** While effective, the protocol-tagging, fast-skip, and queue-based state management are structural, reading as evolutionary within streaming-LLM design rather than algorithmical improvement. Clarifying conceptual novelty beyond integration would help.

**2. Comprehensiveness of the benchmark.** TV-Online consolidates public corpora (Shot2Story, Ego4D, Vript, DiDeMo, Charades) and synthesizes QA via LLMs; many domains (e.g., meetings with ASR lag, robotics, broadcast sports audio) are underrepresented, and audio cues are not addressed.

**3. Evaluator reliance.** The RL stage matches predictions to answers using a binary LLM evaluator within a temporal window—raising bias/robustness questions for reward supervision.

**4. Comparability across settings.** “Online” baselines run at 1 fps under proactive output, while “offline” models receive clips around ground-truth times; heterogeneous interfaces/context budgets complicate strict like-for-like comparisons.

**5. Timestamp sensitivity.** The benchmark assumes accurate triggers and verifies timestamps via a sliding-window scheme; stress-tests for jitter/dropped frames would strengthen claims of temporal robustness.

**6. Throughput realism.** Implementation samples 1 fps with 48 visual tokens on Qwen2.5-VL-3B (8×A800, ~36h); end-to-end latency and behavior at higher fps/edge hardware are unclear.

**7. Synthesis artifacts.** The multi-stage LLM-driven data pipeline (with type-specific checks) may import templated language or coverage biases despite two-stage validation. Auditing artifacts would help.

**8. Missing refereces.** Several influential streaming/online MLLM works from ICLR 2025 and CVPR 2025 are missing from the related work.

**Questions:**

**1. How would synchronized audio (speech/environmental cues) reshape trigger detection, latency, and multi-turn coherence—especially for SQU—given the current vision-focused setup?**

**2. What inter-judge agreement arises between the binary LLM evaluator used in reward shaping and human raters, and can reference-free or pairwise methods reduce bias in semantic matching?**

**3. How do TOM’s fast inference and silent policy perform beyond 1 fps, with longer videos and on-device constraints; can adaptive token budgets preserve temporal precision under tight compute?**

**4. Under noisy triggers or overlapping events, how robust are question/answer IDs and Tag Reward to misbinding—and can uncertainty-aware tagging reduce hallucination penalties?**

**5. Do results transfer across domains not covered by source datasets (e.g., meetings/robotics), and will future releases include diagnostics for demographic or situational bias?**

**6. Which single component (tags, queue state, RL rewards) most drives gains across TGQA/TST/SQU, and can smaller backbones or alternative caches replicate improvements with similar metrics?**

---

> ### Author Response · Authors · 2025-11-28
>
> ### **Reply to Weakness 1**
> We appreciate the reviewer’s acknowledgment of the effectiveness and relevance of our framework. While it is true that components such as protocol-tagging, fast-skip, and queue-based state management draw from established practices in streaming-LLM design, we would like to emphasize that the core conceptual novelty of this work lies not in isolated algorithmic primitives, but in the problem formulation and domain perspective that the community has not previously explored deeply.
>
> Trigger-centric online video understanding is fundamentally different from conventional passive or offline video tasks. As highlighted by Reviewer 2, *“moving from passive, offline video analysis to proactive, trigger-centric online understanding is a crucial and forward-looking step toward more intelligent and interactive systems.”* No existing benchmark, setting, or model has addressed this shift. Our contribution is therefore conceptual at the domain level: identifying the problem, formalizing it, constructing the first high-quality benchmark, and designing a full framework that operationalizes this new paradigm end to end.
>
> We fully agree that algorithmic innovation is important. However, such innovation can only emerge once the underlying task is rigorously defined, its challenges are exposed, and a reliable evaluation protocol exists. Our work establishes a well-defined testbed for future algorithmic advances, rather than mere ad-hoc integrations.
>
> ### **Reply to Weakness 2**
>
> 1. **Underrepresented Domains:** We appreciate the reviewer’s perspective. While benchmark comprehensiveness is indeed desirable, we would like to clarify that this should not be viewed as a weakness for TV-Online at the current stage of the field. Trigger-centric online video understanding is still in its infancy, and even with QA generated from well-established public corpora, we observe that existing models struggle significantly on TV-Online. This indicates that the research gap is substantial, and our goal is precisely to expose and help close this gap by providing the first dedicated benchmark, rather than prematurely covering every possible domain.
> Nevertheless, we find the reviewer’s suggestions highly inspiring. Expanding TV-Online to additional domains (e.g., meetings, robotics, and broadcast sports) would indeed further enrich the benchmark, and we plan to explore these directions in future iterations to continue contributing to the community.
>
> 2. **Audio Cues:** We fully agree that multimodal audio–visual triggers are an exciting future direction. However, current models already perform poorly on the purely visual version of the task; introducing audio at this stage would likely compound the difficulty without offering clear benefits for understanding the core challenges. We therefore focus on the visual domain first, and will incorporate audio-trigger grounding as part of future work once baseline models reach more stable performance.
>
> ### **Reply to Weakness 3**
>
> There are two possible sources of bias: (1) the matching mechanism itself, and (2) the use of an LLM evaluator.
>
> 1. **Bias from the matching mechanism:**
> We intentionally avoid complex reward designs（e.g., such as step-level supervision, dense intermediate rewards, or heuristic rule-based scoring) to minimize reward hacking and reduce structural bias. Instead, we use the most direct and fundamental supervision signal for this task: temporal correctness within a window and binary semantic consistency. These are the minimal, lowest-level, and least bias-prone signals aligned with the final evaluation objective, ensuring that the reward remains both robust and tightly coupled to the task’s end goals.
>
> 2. **Bias from the LLM evaluator (LLM-as-a-Judge):**
> The evaluation performed by the LLM is extremely simple: it only needs to judge semantic equivalence between two short answer phrases (a binary 0/1 decision). This is far easier than open-ended scoring or ranking tasks, and well within the reliability range of modern LLMs. We also deliberately avoid more subjective evaluation forms (e.g., continuous scoring (0–5) or multi-rank comparisons) to further reduce variability and potential bias.) As a result, the binary LLM judging in our setting is both stable and highly robust.

---

> ### Author Response · Authors · 2025-11-28
>
> ### **Reply to Weakness 4**
> We appreciate the reviewer’s concern regarding inconsistent evaluation settings. Importantly, even when offline models are provided with clips centered around the ground-truth timestamps, which effectively gives them partial access to the ground-truth, they still perform poorly. Since they only need to solve the semantic part of the task, their performance remains consistently poor. This indicates that the difficulty of proactive, trigger-centric video understanding is intrinsic and not an artifact of comparable differences. In fact, the offline setting is intentionally more forgiving, and the gap highlights how far current models are from solving this domain, reinforcing the necessity and value of TV-Online.
>
> Moreover, such evaluation protocols are common in other pipeline-based tasks, where benchmarks often assess the latter half of a pipeline under ground-truth conditions for the earlier stages[1][2]. Our offline setup follows this standard practice: it isolates the semantic reasoning ability by removing temporal localization challenges, providing a meaningful and interpretable comparison rather than an unfair one.
>
> [1]:Sabir, Ahmed ,  F. Moreno-Noguer , and Lluís Padró. "Textual Visual Semantic Dataset for Text Spotting." #i{IEEE} (2020).
>
> [2]:Liu, S.; Zhang, X.; Zhang, S.; Wang, H.; Zhang, W. Neural Machine Reading Comprehension: Methods and Trends. Appl. Sci. 2019, 9, 3698.
>
> ### **Reply to Weakness 5**
> Indeed, real-world streaming scenarios may experience temporal jitter or dropped frames, and exploring robustness under such conditions is an important and meaningful direction for future work.
>
> At the same time, we would like to clarify that this aspect should not be viewed as a weakness of TV-Online at the current stage of the field. Existing online video models already struggle considerably even under clean, non-stress conditions; their performance remains low despite the absence of jitter or frame loss. This indicates that the core challenge in proactive, trigger-centric online understanding lies in fundamental temporal reasoning and online decision-making, rather than stress factors associated with noisy inputs. Our benchmark is designed to expose and study these foundational challenges first, before moving toward stress-test scenarios that require substantially stronger temporal robustness.
>
> As future work, we plan to incorporate stress-test modes into TV-Online, providing additional settings to evaluate robustness once baseline models reach more stable performance.

---

> > ### Author Response · Authors · 2025-11-28
> >
> > ### **Reply to Weakness 7**
> > **Semantic Diversity:** Although our data pipeline is LLM-driven, the semantic diversity of TV-Online is primarily determined by the diversity of the underlying video content. Because the videos span a wide range of events, temporal structures, objects, and interactions, the resulting QA pairs naturally exhibit substantial semantic variability, which prevents coverage bias and ensures that the benchmark meaningfully tests video-understanding ability rather than linguistic surface patterns.
> >
> > **Templated Language:**  Regarding the concern about templated language, we agree that LLM-generated questions may share stylistic similarities. However, this does not affect the validity of the benchmark: the task evaluates temporal grounding, trigger detection, and online video reasoning, all of which depend on the video content rather than on linguistic style. In practice, a more consistent question format reduces linguistic noise and allows the benchmark to focus more directly on measuring a model’s ability to understand and respond to the video.
> >
> > ### **Reply to Weakness 8**
> > We sincerely thank the reviewer for the suggestion, and we have added the relevant recent ICLR 2025 and CVPR 2025 streaming MLLM works to the Related Work section.
> >
> > ### **Reply to Question 1**
> > We appreciate the reviewer for raising this interesting and challenging question. Incorporating synchronized audio would indeed reshape several aspects of trigger-centric understanding. Audio may introduce new opportunities (e.g., earlier trigger anticipation through sound onsets, improved multi-turn coherence in SQU via verbal context) but also new challenges, such as cross-modal conflicts when audio and visual cues diverge, modality dominance issues, or increased latency from multimodal alignment and fusion.
> >
> > Addressing these challenges would likely require explicit audio–visual synchronization strategies and robust cross-modal gating mechanisms to handle conflicting cues. Another promising direction is to extend the protocol-tagging scheme so that triggers can be aligned to both audio events and visual events, enabling models to reason jointly over heterogeneous temporal signals. While these directions are beyond the scope of our current vision-only setup, we agree that they open compelling avenues for future research.
> >
> > ### **Reply to Question 2**
> > To address the reviewer’s concern, we compare the binary LLM evaluator with human judgment on a random subset of model–answer pairs. As shown in Table 2, the LLM and human annotators show strong agreement: humans confirm 94/98 LLM-correct cases and 99/102 LLM-incorrect cases. This high consistency indicates that the LLM reliably handles binary semantic matching in our reward setup.
> >
> > **Table 2. Evaluation of Agreement Between LLM Evaluator and Human Raters**
> >
> > | **Results**        | **Human: Correct**    | **Human: Incorrect**     |
> > |--------------------|-----------------------|--------------------------|
> > | **LLM: Correct**   | 94 (Consensus Correct) | 4 (LLM Lenient)           |
> > | **LLM: Incorrect** | 3 (LLM Strict)         | 99 (Consensus Incorrect) |
> >
> > ### **Reply to Question 3:**
> > Please refer to weakness 6.
> >
> > ### **Reply to Question 4:**
> > We appreciate the reviewer for raising this question. In our current benchmark, the trigger–answer pairs are carefully constructed and verified to avoid ambiguous or overlapping events, so misbinding does not occur in the released data.
> >
> > Nevertheless, we agree that noisy triggers and overlapping events are an important and challenging scenario for future exploration. Such cases naturally lead to one-to-many or many-to-many relationships between answers and event IDs, rather than the clean one-to-one bindings assumed in most existing online video understanding tasks. In these settings,  uncertainty-aware or soft tagging strategies may be more appropriate than hard assignments.

---

> > ### Author Response · Authors · 2025-11-28
> > **Reply to Weakness 6 & Question 3**
> >
> > ## **Reply to Weakness 6**
> >
> > ### **Performance under Inference Constraints :**
> > We appreciate the insightful question regarding inference dynamics and resource constraints. To address this, we conducted two new ablation studies during the rebuttal, varying the frame sampling rate (FPS) and the visual token budget. The results are summarized in Table 1.
> > - **Robustness beyond 1 FPS (FPS Scaling)**: We evaluated the model at 0.5, 1.0, and 2.0 fps. The results demonstrate that our method remains robust to higher sampling rates.
> >   - **Trade-off Dynamics**: According to Table 1, as the frame sampling rate increases, the model observes more frames within the fixed memory window $W$. This leads to a slight increase in Recall (more opportunities to capture events) but a minor decrease in Precision (more active responses). Consequently, the Overall F1 remains stable.
> >   - **Robustness via FIM**: The Fast Inference Mechanism (FIM) plays a key role here. By detecting and skipping highly redundant frames (which naturally increase at higher FPS), FIM ensures that the computational load and model behavior remain stable. This confirms that our design is robust to varying sampling rates and does not over-fit to the 1 fps setting.
> >
> > - **Feasibility under Tight Compute** :  we directly reduced the visual tokens to 24 per frame without retraining the model. As shown in Table 1, while there is a slight performance decrease compared to the full-token baseline, the model maintains a respectable F1 score and remains fully functional.This robustness is attributed to two factors:
> >   - **Backbone Capability**: The strong generalization of the Qwen2.5-VL backbone allows it to interpret compressed visual representations effectively.
> >   - **Task Nature**: The TV-Online task prioritizes high-level temporal logic over fine-grained spatial details. Thus, the model can still capture core events even with fewer tokens, validating the potential for adaptive budgets in on-device scenarios.
> >
> > **Table 1: Impact of different frame sampling rates (FPS) and visual tokens on model performance.**
> >
> > |**FPS/Visual tokens Setting**|**Precision**|**Recall**|**OverallF1**|
> > |:------------------------:|:-------------:|:----------:|:--------------:|
> > |**0.5/48**|43.1(+1.1)|31.2(-2.6)|36.2(-1.3)|
> > |**1.0/48(Default)**|42.0|33.8|37.5|
> > |**2.0/48**|40.3(-1.7)|35.5(+1.7)|37.7(+0.2)|
> > |**1.0/24**|40.7(-1.3)|32.5(-1.3)|36.1(-1.4)|
> >
> > ### **Discussion on On-device Deployment and Latency**
> > We acknowledge the importance of end-to-end latency and on-device constraints for real-world applications. While specific hardware-level optimization (e.g., quantization, mobile inference) lies somewhat outside the current scope of our algorithmic contributions, we emphasize that our framework is designed with extensibility in mind. Future work could incorporate adaptive token budgets, hierarchical caching, or dynamic frame selection to better support higher throughput and resource-constrained deployment.

---

> ### Author Response · Authors · 2025-11-28
>
> ### **Reply to Question 5**
> **Benchmark Generalization:**
> We thank the reviewer for raising this important question. Although TV-Online is built from a specific set of video sources, our experiments already show that TOM generalizes beyond these domains, it achieves **state-of-the-art performance (37.35)** on ProactiveVideoQA[1] and remains **competitive (2.30)** on Vispeak-bench[2], despite the latter relying on audio-heavy cues absent from our training data. This indicates that the proposed framework captures abilities that transfer across domain boundaries, even without domain-specific tuning. More detailed comparisons, including those between our model and other benchmarks, will be included in the revised version.
> Meanwhile, we fully agree that expanding to additional domains (e.g., meetings, robotics, or audio-augmented video) would further strengthen the benchmark, and we plan to incorporate such sources in future releases.
>
> **Demographic or Situational Bias:**
> We will provide distributional statistics of the current dataset in the revised version and are committed to adding more explicit diagnostic tools in future versions to help the community better understand potential bias patterns, Some details of our training dataset have been shown in the Table 4.
>
> **Table 4: Macro-Domain and Category Distribution in TV-Online**
>
> | **Macro-Domain**| **Detailed Categories Covered** | **Primary Data Sources** |
> |:-----------------------------------:|:-----------------------------------------------------------------------:|:------------------------------:|
> | **1. Daily Life & Household** | Baking, Laundry, Cooking, Cleaning, Knitting, Indoor Object Interaction | Charades, Ego4d, Vript |
> | **2. Social, Travel & Lifestyle** | Vlogs, Travel, Pets, Autos, Casual Events, Family Moments | DiDeMo, Shot2Story, Vript|
> | **3. Arts, Entertainment & Gaming** | Film, Animation, Comedy, Gaming, Reading| Shot2Story, Vript|
> | **4. Knowledge, Science & News**| Science, Technology, Education, Politics| Shot2Story, Vript|
> | **5. Sports & Action**| Professional Sports, Casual Athletics, High-motion activities | All Datasets |
>
> ### **Reply to Question 6**
> **ID Tagging is the primary driver for performance gains:**
> As shown in our comprehensive ablation study (Section 5.4), the ID tagging mechanism serves as the most critical component. According to Table 7 of the main paper, removing ID tagging results in a significant drop of *11.8%* in SQU and *6.8%* in Overall F1. This indicates that explicitly tracking object identities is fundamental for the model to handle the TST and SQU tasks effectively. ID tagging substantially improves TOM’s ability to preserve entity consistency over time. In our evaluations, models without ID tagging frequently conflate different entities with similar appearances, lose track of previously grounded objects, or produce responses whose referents drift across turns. These errors accumulate in long-horizon settings and lead to large performance drops in tasks such as SQU, where multi-turn reasoning requires stable entity continuity. By contrast, incorporating ID tagging markedly reduces these failure cases, yielding more coherent multi-turn predictions and significantly enhancing sequential reasoning accuracy.
>
> **Replicability on different backbones and alternative caches:**
> Regarding your inquiry about smaller backbones, we would like to clarify that our main model utilizes a 3B backbone(Qwen2.5-VL-3B),which is already a compact and efficient model compared to standard 7B baselines. To fully address your concern about whether the improvements can be replicated across different backbone scales. We also trained a larger 7B variant (ToM-7B) under the same settings. As shown in Table 4 and Table 5, applying our method to the 7B backbone yields consistent gains. This confirms that our proposed components (ID Tags, TSM) are robust and drive improvements in the different backbone.
>
> **Table 4: Ablation Study of Tags**
>
> |**Method**|**TGQA\(F1\)**|**TST\(F1\)**|**SQU\(F1\)**|**Overall\(F1\)**|
> |:---------------------:|:--------------:|:-------------:|:-------------:|:-----------------:|
> |**TOM\-7B\(w/oIDTags\)**|51|35\.2|30\.5|38\.6|
> |**TOM\-7B\(w/IDTags\)**|53\.8|38\.5|36\.7|43\.1|
> |**PerformanceGain**|+2\.8|+3\.3|+6\.2|+4\.2|
>
> **Table 5: Ablation Study of TSM**
>
> |**Method**|**SQU\(F1\)**|**Overall\(F1\)**|
> |:------------------:|:-------------:|:-----------------:|
> |**TOM\-7B\(w/oTSM\)**|34\.1|41\.9|
> |**TOM\-7B\(w/TSM\)**|36\.7|43\.1|
> |**PerformanceGain**|2\.6|1.2|
>
>
> [1] Wang Y, Meng X, Wang Y, et al. Proactivevideoqa: A comprehensive benchmark evaluating proactive interactions in video large language models. arXiv preprint arXiv:2507.09313, 2025.
>
> [2] Fu, Shenghao, et al. "ViSpeak: Visual Instruction Feedback in Streaming Videos." arXiv preprint arXiv:2503.12769 (2025).

---

### Official Review · Reviewer_eJP8 · 2025-10-30

**Soundness:** 3
**Presentation:** 2
**Contribution:** 3
**Rating:** 4
**Confidence:** 3

**Summary:**

The paper introduces TV-Online, a large-scale dataset for online video understanding that covers three tasks: Temporal Grounded Question Answering (TGQA), Temporal State Tracking (TST), and Sequential Query Understanding (SQU). The proposed model incorporates two key modules, Fast Inference Mechanism and Trigger State Management, to enable efficient streaming and proactive response generation. Experiments on TV-Online, along with ablation studies, demonstrate the effectiveness of the proposed approach.

**Strengths:**

- **Scale and diversity:** TV-Online is a large and diverse benchmark, covering multiple aspects of online video understanding.
- **Model effectiveness:** The proposed method outperforms prior online models and is supported by detailed ablation studies.

**Weaknesses:**

- In Section 4.1, the modeling details are insufficient. The descriptions of the Fast Inference Mechanism and Trigger State Management lack clarity, making it difficult to fully understand how the method works. For example, what constitutes a "lightweight off-the-shelf model", and how does it analyze both inputs and previous outputs? More architectural details are needed.

- Correspondingly, the training process is underexplained. How many videos and instructions are used for each of the three training stages? How does the scale of the training data influence the performance?

- I would not agree with the claim in L419–421 that TOM surpasses all online and most offline models. In TGQA, TOM significantly underperforms compared to several offline and streaming models. Have the authors trained larger (e.g., 7B) variants for fair comparison?

- While TOM outperforms existing online models on TV-Online, I would recommend evaluating it on established streaming video understanding benchmarks (e.g., OVO-Bench [1], Streaming-Bench [2]) to assess generalization.

**References**

[1] OVO-Bench: How Far is Your Video-LLMs from Real-World Online Video Understanding?, CVPR 2025

[2] StreamingBench: Assessing the Gap for MLLMs to Achieve Streaming Video Understanding, arXiv 2024

**Questions:**

See the weaknesses

---

> ### Author Response · Authors · 2025-11-28
> **Reply to Weakness 1 & Weakness 2**
>
> ## **Reply to Weakness 1: Clarifications on Fast Inference and TSM**
>
> We apologize for the lack of clarity in Section 4.1. We will revise the manuscript to provide precise modeling details on the following two key components:
> ### 1. **Fast Inference Mechanism:**
> To ensure low latency in fully online streaming, we utilize an HSV (Hue, Saturation, Value) color histogram comparison to measure the visual similarity between the current frame and the previous frame. This lightweight metric allows for rapid computation. If the histogram similarity score exceeds a pre-defined threshold $\tau$, the system identifies the frame as redundant and skips detailed processing.
> - Skip Logic: For these redundant frames, we bypass the model’s forward pass (defaulting to a `<|silent|>` state) to save computation, while still retaining the visual token in the context window to maintain temporal continuity.
> - Active Inference: Only when the similarity is below $\tau$ do we perform a forward pass to calculate the probabilities of `<|response|>` vs. `<|silent|>`. This significantly reduces latency compared to processing every frame.
>
> ### 2. **Trigger State Management**
> - **Specific Model Identity:** The "lightweight off-the-shelf model" refers to Qwen3-1.7B. We selected this specific model primarily to minimize deployment costs and inference latency. However, our framework is agnostic to the text backbone; this module can be flexibly replaced by other off-the-shelf text-only LLMs depending on the available computational resources.
>
> - **Motivation:** Standard MLLMs struggle in online streaming scenarios where continuous video frames and Q&A pairs are interleaved. As highlighted in recent studies[1,6,7], processing interleaved visual-text streams in multi-turn dialogues poses a significant challenge. The massive influx of visual information competes with textual history, causing the model to lose track of user intent in multi-turn interactions. However, analyzing dialogue history to resolve dependencies is a trivial task for pure text LLMs. Therefore, we decouple this logic, using a lightweight text model to ensure robust intent understanding without the interference of visual redundancy.
>
> - **Event-Driven Mechanism:** This module implements the Trigger State Management as detailed in our paper. Specifically, we maintain a chronologically ordered structured queue that records the sequence of User Query and Model Response pairs. Upon a new trigger, the model analyzes its relevance to this history to identify dependencies, contextual links, or contradictions. The derived relational information is used to rewrite and enrich the current query for the main backbone. For instance, in a sports scenario where the history notes `A1: The blue player scored a goal` and the user subsequently asks `Q2: What did this player do?`, the TSM module resolves the dependency and outputs the enriched query:
> `What did the blue player do?`. This enables the main model to make context-aware predictions that remain consistent across interactions.
>
> We will clarify these mechanisms in Section 4.1 and add an appendix section that includes more implementation details.
>
>
>
> ## **Reply to Weakness 2: Elaboration on Training Data Composition and Scaling Analysis**
>
> We thank the reviewer for pointing out the need for more details regarding our training pipeline. We have organized the statistics of our three-stage training process and analyzed the impact of training scale/progress as follows:
>
> (1) **Training Data Statistics**: Table 1 details the exact composition of videos and instructions used in each stage. Our pipeline progresses from large-scale alignment (Stage 1) to supervised fine-tuning (Stage 2), and finally to reinforcement learning on a high-quality subset (Stage 3).
>
> **Table 1: Statistics of training data across three stages**
>
> | **Stage** | **Data Sources (Videos)** | **Text Statistics** |
> | :--- | :--- | :--- |
> | **Stage 1: Pre-training** | • Shot2Story: 27.8k • Vript: 30.7k | • Instructions: 58k • Answers: 338k |
> | **Stage 2: SFT**          | • TV-online (Train): 48k | • Instructions: 256k • Answers: 500k          |
> | **Stage 3: RL**           | • TV-online (Subset): 1k* | • Instructions: 1.8k • Answers: 3.0k         |
>
>
> (2) **Impact of Training Scale**: To demonstrate how the scale of training influences performance, we analyzed the model's performance at different training steps during the SFT stage. As shown in Table 2, the model exhibits consistent performance gains as training progresses and more data samples are consumed, confirming that our model effectively benefits from the scale of the dataset.
>
> **Table 2: Ablation study on the impact of training data scale.**
>
> | **Training Steps** | **TGQA** | **TST** | **SQU** |
> |:--------------------:|:--------:|:-------:|:-------:|
> | **3000 steps**       | 40.1     | 24.1    | 24.8    |
> | **5000 steps**       | 43.6     | 28.3    | 28.5    |
> | **Final**            | 44.5     | 30.0      | 29.8    |

---

> > ### Author Response · Authors · 2025-11-28
> > **Reply to Weakness 3 & 4**
> >
> > ## **Reply to Weakness 3: Clarification on Performance Claims & ToM-7B Evaluation**
> >
> > We appreciate the reviewer’s constructive feedback regarding our claims. We agree that the statement in L419–421 was overly broad, and we would like to clarify our position as follows:
> > 1. **Refined Scope**: We will revise the manuscript to strictly clarify that ToM surpasses all online models and equivalent-sized offline models.
> > 2. **Protocol-Induced Gap**: The TGQA gap stems from distinct protocols: offline models benefit from ground-truth clips (serving as a semi-theoretical upper bound), whereas ToM must autonomously determine timing and content from continuous streams. This unconstrained setting is inherently more challenging than pre-segmented inference.
> > 3. **Fair Comparison**: Despite these setting differences, to address the concern regarding model capacity and ensure a fair comparison, we trained a larger 7B variant (ToM-7B). Due to time and resource constraints during the rebuttal, ToM-7B was trained using LoRA (with only embedding/head layers unfrozen). As shown in Table R3, even with this constrained setup, ToM-7B achieves 53.8% on TGQA (+8.6% over 3B) and an Overall score of 43.1%. This demonstrates that when aligned with comparable model sizes, ToM outperforms existing online models and narrows the gap with offline approaches, even under stricter streaming settings.
> >
> > **Table 3: Performance comparison between 3B and 7B variants.**
> >
> > | **Model**  | **TGQA** | **TST** | **SQU** | **Overall** |
> > |:----------:|:--------:|:-------:|:-------:|:-----------:|
> > | **ToM-3B** | 45.2        | 35.0       | 31.8       | 37.5        |
> > | **ToM-7B** | 53.8(+8.6%) | 38.5(+3.5%)| 36.7(+4.9%)| 43.1(+5.6%) |
> >
> >
> >
> >
> > ## **Reply to Weakness 4: Extensive Generalization Evaluation on Established Benchmarks**
> >
> > We fully agree with the reviewer on the importance of evaluating generalization on established benchmarks. We have now conducted extensive additional evaluations. Specifically, we not only tested on the suggested OVO-Bench[1] and Streaming-Bench[2] but also extended our evaluation to include Vispeak-Bench[3], ProactiveVideoQA[4], OmniMMI[5], and Svbench[6].We will include the full experimental settings and detailed results in the revised manuscript.
> >
> > **Generalization on StreamingBench & OvO Benchmarks**
> > For fairness and to control for model capacity, we also trained a larger 7B variant (ToM-7B) under the same settings. As shown in Table 4, ToM-7B demonstrates superior generalization. For StreamingBench, ToM achieves a SOTA Overall Score of 48.78, significantly outperforming baselines like Dispider (45.63) and Qwen2.5-VL (46.08). Notably, it leads in Contextual Understanding (34.2), validating our model's ability to maintain long-term context.For OvOBench, In Forward Active Responding, ToM achieves 54.68%, surpassing the all other models. This confirms that our Event-Driven mechanism effectively enables precise, real-time proactive interaction.This superiority confirms the robust generalization capability of our model, demonstrating that it can effectively adapt to diverse, open-ended streaming scenarios.
> >
> >
> > **Table 4: Experimental Results on StreamingBench and OvO-Bench**
> >
> > |**Benchmarks**|||**Streamingbench**|||**OvObech**|
> > |:--------:|:--------:|:--------:|:--------:|:--------:|:--------:|:--------:|
> > |**Method**|Frames|Real-TimeVisualUnderstanding|OmniSourceUnderstanding|ContextualUnderstanding|Overall|ForwardActiveResponding|
> > |**VideoLLM-Online**[9]|2fps|35.99|28.45|26.55|30.33|-|
> > |**FlashVStream**[13]|1fps|23.23|26|24.12|24.45|44.23|
> > |**Dispider**[8]|1fps|67.63|35.66|33.61|45.63|34.72|
> > |**ViSpeak(s2)**[3]|1fps|74.36|-|-|-|54.25|
> > |**Timechat-Online**[10]|1fps|75.36|-|-|-|36.70|
> > |**StreamBridge**[12]|1fps|77.04|-|-|-|48.36|
> > |**StreamForest**[11]|1fps|**77.26**|-|-|-|53.49|
> > |**Qwen2.5VL-7B**[14]|1/0.5/0.2fps|73.76|35.2|29.3|46.08|40.27|
> > |**ToM-7B(ours)**|1/0.5/0.2fps|74.72|**37.3**|**34.2**|**48.78**|**54.68**|

---

> > > ### Author Response · Authors · 2025-11-28
> > > **References**
> > >
> > > ## **References**
> > >
> > > [1] Lin, Junming, et al. "Streamingbench: Assessing the gap for mllms to achieve streaming video understanding." arXiv preprint arXiv:2411.03628 (2024).
> > >
> > > [2] Niu, Junbo, et al. "OVO-Bench: How Far is Your Video-LLMs from Real-World Online Video Understanding?." Proceedings of the Computer Vision and Pattern Recognition Conference. 2025.
> > >
> > > [3] Fu, Shenghao, et al. "ViSpeak: Visual Instruction Feedback in Streaming Videos." arXiv preprint arXiv:2503.12769 (2025).
> > >
> > > [4] Wang Y, Meng X, Wang Y, et al. Proactivevideoqa: A comprehensive benchmark evaluating proactive interactions in video large language models. arXiv preprint arXiv:2507.09313, 2025.
> > >
> > > [5] Wang, Yuxuan, et al. "OmniMMI: A Comprehensive Multi-modal Interaction Benchmark in Streaming Video Contexts." Proceedings of the Computer Vision and Pattern Recognition Conference. 2025.
> > >
> > > [6] Yang, Zhenyu, et al. "Svbench: A benchmark with temporal multi-turn dialogues for streaming video understanding." arXiv preprint arXiv:2502.10810 (2025).
> > >
> > > [7] Liu, Ziyu, et al. "Mmdu: A multi-turn multi-image dialog understanding benchmark and instruction-tuning dataset for lvlms." Advances in Neural Information Processing Systems 37 (2024): 8698-8733.
> > >
> > > [8] Qian, Rui, et al. "Dispider: Enabling video llms with active real-time interaction via disentangled perception, decision, and reaction." Proceedings of the Computer Vision and Pattern Recognition Conference. 2025.
> > >
> > > [9] Chen, Joya, et al. "Videollm-online: Online video large language model for streaming video." Proceedings of the IEEE/CVF Conference on Computer Vision and Pattern Recognition. 2024.
> > >
> > > [10] Yao, Linli, et al. "Timechat-online: 80% visual tokens are naturally redundant in streaming videos." Proceedings of the 33rd ACM International Conference on Multimedia. 2025.
> > >
> > > [11] Zeng, Xiangyu, et al. "StreamForest: Efficient Online Video Understanding with Persistent Event Memory." arXiv preprint arXiv:2509.24871 (2025).
> > >
> > > [12] Wang, Haibo, et al. "StreamBridge: Turning Your Offline Video Large Language Model into a Proactive Streaming Assistant." arXiv preprint arXiv:2505.05467 (2025).
> > >
> > > [13] Zhang, Haoji, et al. "Flash-vstream: Memory-based real-time understanding for long video streams." arXiv preprint arXiv:2406.08085 (2024).
> > >
> > > [14] Bai, Shuai, et al. "Qwen2. 5-vl technical report." arXiv preprint arXiv:2502.13923 (2025).

---

### Official Review · Reviewer_wxSm · 2025-10-31

**Soundness:** 3
**Presentation:** 3
**Contribution:** 3
**Rating:** 6
**Confidence:** 4

**Summary:**

This paper introduces a new benchmark and framework for a critical and practical task: trigger-centric online video understanding. The authors argue that existing systems for online video analysis lack the proactive, real-time responsiveness required for many applications. To address this, they make two primary contributions. First, they introduce TV-Online, a large-scale dataset featuring 50K videos and 500K time-stamped answers, specifically designed to evaluate trigger-based tasks. The dataset is hierarchically structured into progressively complex tasks, from basic temporal grounding to multi-turn, asynchronous reasoning. Second, they propose TOM (Trigger-Oriented Model), a streaming-oriented framework that employs an agent-like paradigm. TOM uses special protocol tags for response control, a queue-based state management system for handling multi-turn context, and a progressive training strategy that culminates in reinforcement learning to refine the model's responsiveness, coverage, and coherence. The paper presents extensive experiments demonstrating that TOM significantly outperforms existing online methods and establishes a strong baseline on the new TV-Online benchmark.

**Strengths:**

- The paper tackles a highly relevant and forward-looking problem. Moving from passive, offline video analysis to proactive, trigger-centric online understanding is a crucial step towards building more intelligent and interactive systems. The problem formulation is clear and well-motivated.
- The proposed TV-Online dataset is a major contribution in itself. Its large scale, diverse tasks, and hierarchical structure of difficulty (from Basic to Expert) provide a much-needed, systematic way to measure progress in this area. The comparison in Table 1 effectively highlights the gap it fills compared to existing datasets.
- The proposed TOM framework is thoughtfully designed to address the specific challenges of the task. The combination of a streaming-aware architecture, explicit state management for multi-turn dialogue, and a progressive three-stage training pipeline (pre-training, fine-tuning, RL) is holistic and sound. The use of reinforcement learning with custom rewards to optimize temporal accuracy and coverage is particularly fitting.
- The experimental evaluation is comprehensive. The authors compare their model against a wide range of both offline and online baselines, demonstrating significant performance gains. The ablation studies (Tables 4, 5, 7) are particularly valuable as they clearly validate the effectiveness of each component of the proposed framework, such as the progressive training stages and the ID tagging protocol.

**Weaknesses:**

- While the overall framework and its application to this specific problem are novel and effective, many of the individual technical components (e.g., using special tokens for control, progressive training, RL for policy alignment) are adaptations of existing techniques in the broader field of large language and multimodal models. The main novelty lies in their synergistic combination and customization for the trigger-centric video task, rather than a fundamental algorithmic breakthrough.
- The paper's goal is to advance "real-time" video intelligence. It introduces a "Fast Inference Mechanism" that skips redundant frames based on visual similarity. However, the analysis of this mechanism is limited. A more in-depth discussion on its computational overhead, actual latency reduction (e.g., in seconds or FPS), and the potential trade-offs (e.g., how the similarity threshold affects the risk of missing subtle but important trigger events) would be crucial to substantiate the "real-time" claims.
- The data generation pipeline relies heavily on LLMs to synthesize and verify QA pairs. While this is a common and scalable approach, it can introduce artifacts or biases from the generator models. The paper mentions a two-stage validation process, but further details on the extent of human verification, inter-annotator agreement (if applicable), and strategies used to mitigate potential LLM-induced biases would strengthen the credibility and quality of the TV-Online benchmark.

**Questions:**

- Could you provide more details on the "Fast Inference Mechanism"? Specifically, how is the visual similarity threshold determined and tuned? Have you analyzed the performance impact (both in terms of F1-score and latency) under different threshold settings? What are the failure modes, i.e., could this mechanism cause the model to miss triggers that occur during visually static but semantically important moments?
- Regarding the dataset construction: What was the scale of human involvement in the two-stage validation process? Were there quantitative metrics (e.g., acceptance/rejection rate, human evaluation scores) used to ensure the quality and correctness of the LLM-generated questions and time-stamped answers?
- In the post-training RL stage, the final reward is an aggregation of RFβ and Rtag with weights α and γ. Could you provide some intuition or results from a sensitivity analysis on how these hyperparameters were chosen and how they affect the model's final behavior (e.g., the trade-off between being responsive and remaining silent)?

---

> ### Author Response · Authors · 2025-11-28
>
> We sincerely thank the reviewer for your thoughtful and detailed feedback. We appreciate that you find our work valuable and recognize the contributions it makes to the field of real-time video understanding. We are pleased to hear that the proposed framework aligns with the forward-looking goals of advancing trigger-centric online understanding. Below, we provide a point-by-point response to address your questions and clarify the concerns raised.
> ### **Reply to Weakness 1**
> We appreciate the reviewer’s thoughtful observation. While it is true that TOM draws on established modeling ingredients such as control tokens, progressive training, and reinforcement learning, the core contribution of this work lies in defining and operationalizing a new problem setting rather than introducing a single isolated algorithmic primitive. Trigger-centric online video understanding has remained largely unexplored, and existing techniques are not directly applicable without substantial task-driven redesign. Our framework provides a principled formulation of the task, a unified protocol for handling temporal triggers and multi-turn online reasoning, and a comprehensive benchmark that enables systematic evaluation across TGQA, TST, and SQU.
>
> From an academic perspective, we see this as an infrastructural contribution: it establishes the foundations, data resources, and methodological interface needed for future algorithmic innovation in this domain. By introducing a standardized evaluation environment and demonstrating a full-stack solution tailored to the nuances of proactive online reasoning, the work aims to support and catalyze research in an area where the community currently lacks established tools or baselines. We appreciate the reviewer’s comment, and we will clarify this positioning more explicitly in the revision.

---

> ### Author Response · Authors · 2025-11-28
>
> ### **Reply to Weakness 2 & Question 1**
> 1. **Mechanism Details & Threshold Tuning:**
> To ensure low latency in fully online streaming, we utilize an HSV (Hue, Saturation, Value) color histogram comparison to measure the visual similarity between the current frame and the previous frame. This lightweight metric allows for rapid computation. If the histogram similarity score exceeds a pre-defined threshold $\tau$, the system identifies the frame as redundant and skips detailed processing.
> - Skip Logic: For these redundant frames, we bypass the model’s forward pass (defaulting to a <|silent|> state) to save computation, while still retaining the visual token in the context window to maintain temporal continuity.
> - Active Inference: Only when the similarity is below $\tau$ do we perform a forward pass to calculate the probabilities of <|response|> vs. <|silent|>. This significantly reduces latency compared to processing every frame.
> - Threshold Tuning: The similarity threshold ($\tau$) was determined empirically by evaluating a range of candidate values on our validation set. We aimed to identify a critical balance point where inference latency drops while the F1-score remains stable. As shown in Table 1, our selected threshold represents the optimal trade-off, minimizing computational overhead with negligible accuracy loss.
> 2. **Performance Trade-off Analysis:**
> Table 1 summarizes the trade-offs introduced by the Fast Inference Mechanism (FIM) under different similarity thresholds ($\tau$). As$\tau$decreases, the model skips more redundant frames, leading to significantly reduced latency and higher throughput. With our setting of$\tau$= 0.98, FIM increasing throughput from 2.36 fps to 3.08 fps, with only minimal performance degradation(−0.3% in F1 and −0.8% in recall). While a more aggressive threshold ($\tau$=0.92) offers higher speed (up to 5.16 fps), it results in noticeable accuracy drops(-3.6% in F1 and -5.2% in recall), emphasizing the importance of finding a balanced threshold. These results demonstrate that FIM significantly improves real-time efficiency with negligible impact on task performance. These findings were obtained on the TV-Online benchmark.
>
> **Table 1. Trade-offs of Fast Inference Mechanism**
> |Threshold($\tau$)|Throughput(FPS)|OverallF1(%)|Recall(%)|Precision(%)|
> |----------------------|------------------|----------------|--------------------|--------------------|
> |w/o FIM|2.36|37.5|33.8|42.2|
> |$\tau$=0.98|3.08(+0.72)|37.2(-0.3)|33.0(-0.8)|42.8(+0.6)|
> |$\tau$=0.95|3.81(+1.45)|36.7(-0.8)|31.8(-2.0)|43.4(+1.2)|
> |$\tau$=0.92|5.16(+2.80)|33.9(-3.6)|28.6(-5.2)|44.5(+2.3)|
>
> 3. **Failure Modes & Mitigation:**
> Regarding the failure modes, we agree that the Fast Inference Mechanism may miss triggers that occur during visually static but semantically important moments. This is a natural limitation of using HSV-based visual similarity for redundancy detection: when the frames are nearly identical, FIM intentionally skips them to reduce latency. Importantly, such cases often rely on non-visual cues (e.g., speech, acoustic reactions, or environmental sounds) which are outside the scope of our vision-only setup. Indeed, benchmarks such as ViSpeak[1] show that audio plays a crucial role in categories like Anomaly Warning (AW) and Humor Reaction (HR), where the visual stream remains almost static. We view multimodal extensions, particularly incorporating audio, as a promising direction to mitigate this limitation.
>
> [1] Fu, Shenghao, et al. "ViSpeak: Visual Instruction Feedback in Streaming Videos." arXiv preprint arXiv:2503.12769 (2025).

---

> ### Author Response · Authors · 2025-11-28
>
> ### **Reply to Weakness 3 & Question 2**
> **Quantitative Quality Assurance on Test Set:**
> Our data construction primarily relies on a two-stage automated multi-dimensional validation framework, without human validation during the construction phase to ensure scalability. However, considering the importance of data quality for task evaluation, we perform a human verification of the test set: three annotators with relevant background independently review 500 randomly selected samples, assessing the clarity of questions, relevance to video content, and accuracy of timestamped answers. The results show an overall acceptance rate of 94.2%, with an inter-annotator agreement (Fleiss' Kappa) of 0.86, indicating high accuracy and reliability of the generated data. This verification strongly supports the quality of our dataset, and we will include the details in the paper's appendix for reference.
>
> **Two-Stage Filtering Mechanism:**
> Regarding the data construction process: In the first stage, we construct a hierarchical content quality validation system for the tasks of TGQA, TST, and SQU, which increase in complexity. By using a large language model (LLM-as-a-Judge), we score the samples on a 0–10 scale: Basic level (TGQA): Ensures that the question has the "online trigger" property (i.e., serves as a monitoring command for future events), is semantically clear, has no answer leakage, and that the answer accurately comes from the original annotation; Intermediate level (TST): In addition to the basic checks, we further validate whether the question focuses on continuous tracking of objects/attributes and check the completeness of the timestamp coverage; Advanced level (SQU): Requires further context coherence between multiple-turn questions, ensuring that follow-up questions are valid and based on previous answers. Only high-confidence samples are retained (i.e., with scores >7 for the training set and ≥9 for the test set) . In the second stage, we use an offline multimodal large language model to validate timestamp alignment through a sliding window strategy: the model only has access to the historical video and dialogue context up to the current moment, and the timestamp is marked as correct when the model is first confident in its answer. This mechanism effectively prevents future information leakage and significantly alleviates "temporal hallucinations." To ensure transparency and reproducibility, we have made all validation prompt templates and sliding window parameters (such as window length and step size) publicly available in the appendix. This automated framework not only avoids the bottleneck of human validation but also effectively controls generation bias.

---

> > ### Author Response · Authors · 2025-11-28
> >
> > ### **Reply to Question 3**
> > **Sensitivity Analysis: Decoupling Responsiveness from Alignment:**
> > In the post-training RL stage, the final reward is an aggregation of$𝑅_{𝐹_{\beta}}$and$𝑅_{tag}$weighted by$\alpha$ and$\gamma$, respectively. The purpose of $R_{F_{\beta}}$ is to balance both accuracy and completeness of the answers, as it combines precision and recall. The internal$\beta$ parameter controls the model's preference between these two: larger $\beta$ values emphasize recall, encouraging the model to cover all correct answers and thus be more "responsive." Smaller $\beta$ values prioritize precision, making the model output answers only when highly confident, leading to a tendency to "remain silent."
> >
> > The $𝑅_{tag}$ term ensures that the answers generated by the model correspond clearly and directly to the input question, penalizing outputs that fail to align. This mechanism helps suppress hallucinations and unsubstantiated answers, indirectly encouraging more conservative behavior. As shown in Table 2, While 𝛼 and $\gamma$ control the relative importance of the two reward terms, they do not directly govern the model's responsiveness or silence. Our experiments show that the real driver of this trade-off is the $\beta$ parameter in $R_{F_{\beta}}$. As shown in Table 8 of the main paper (e.g., Ablation Study), increasing $\beta$ significantly increases the number of answers produced by the model, reflecting stronger "responsive" behavior, while decreasing $\beta$ leads to more cautious and silent behavior. In practice, we primarily adjust $\beta$ to control the model's responsiveness, while $\alpha$ and $\gamma$ are used to balance the trade-off between coverage and alignment reliability in the overall optimization goal.
> >
> > **Hyperparameter Selection:** Rationale for Small$\gamma$Weight
> > In our design, we fixed $\alpha=1$ for $R_{F_\beta}$ as the primary objective because our core goal is to optimize the structural quality (precision/recall trade-off) of the generated stream.However, relying solely on $R_{F_\beta}$ can be problematic.$R_{F_\beta}$ can sometimes be coarse-grained. For instance, it may assign similarly low rewards to two very different failure modes: (a) correct tag but poor explanation, and (b) incorrect tag. To address this, we introduced $R_{tag}$ as an auxiliary supervision signal. We set $\gamma=0.2$ (a relatively small weight) to specifically enforce the correctness of the high-level decision. This ensures the model learns to predict the correct tag, while keeping the primary training focus on improving the quality of the generated explanation via $R_{F_\beta}$.
> >
> > **Table 2: Ablation study on the impact of reward weights**
> > |**$\alpha$**|**γ\($R\_{tag}$Weight\)**|**Slient Rate\(%\)**|
> > |:-----:|:------------------------:|:---------------------:|
> > |1|0|82\.3|
> > |1|0\.2|82\.4|
> > |1|0\.4|82|

---

### Official Review · Reviewer_dPUS · 2025-11-01

**Soundness:** 1
**Presentation:** 2
**Contribution:** 2
**Rating:** 4
**Confidence:** 5

**Summary:**

To address the gap in online video understanding, the paper proposes TV-Online (Trigger Video-Online), a large-scale dataset consisting of 50K videos, 200K questions, and 500K time-stamped answers. This dataset encompasses progressively complex trigger-based tasks such as temporal grounding, asynchronous scheduling, and multi-trigger reasoning. To leverage this dataset, the paper introduces a streaming-oriented model (TOM) that utilizes protocol-level tagging and structured state management, enabling frame-by-frame decision-making with precise timing and consistent responses to asynchronous, multi-turn triggers.

**Strengths:**

- The paper introduces the TV-Online dataset, which is large in scale, consisting of 50K videos, 200K questions, and 500K time-stamped answers.

- The proposed state management mechanism is interesting .

**Weaknesses:**

- The paper fails to clearly explain how the TV-Online dataset is divided into training and testing sets. Furthermore, Table 1 mixes the training data with benchmark comparisons, which is highly unfair.

- On the TV-Online evaluation set, how is the accuracy of the answers ensured, and how is the difficulty of the questions controlled?

- The proposed TOM is only tested on the TV-Online evaluation set, which does not sufficiently validate the effectiveness. How does it perform on other benchmarks designed for proactive video understanding, such as Vispeak[1] and ProactiveVideoQA[2]?

[1] Fu S, Yang Q, Li Y M, et al. ViSpeak: Visual Instruction Feedback in Streaming Videos[J]. arXiv preprint arXiv:2503.12769, 2025.

[2] Wang Y, Meng X, Wang Y, et al. Proactivevideoqa: A comprehensive benchmark evaluating proactive interactions in video large language models[J]. arXiv preprint arXiv:2507.09313, 2025.

**Questions:**

See weaknesses above

---

> ### Author Response · Authors · 2025-11-27
>
> We thank the reviewer for the constructive and insightful comments. We have carefully revised the manuscript and improved the clarity of several sections in response to all concerns. Below, we provide a point-by-point reply.
> ### **Reply to Weakness 1**
> We have clarified and refined the description of the training–testing split to make it fully transparent. The final split guarantees no overlap between the training and test sets, and all training videos come exclusively from the training split of the original dataset, without including any testing videos from other benchmarks. The details of the benchmark are shown in Table 1.
>
> **Table 1. Statistics of the TV-Online Evaluation Set**
>
> | Task         | Videos | Questions | Answers |
> |--------------|--------|-----------|---------|
> | Basic        | 1150   | 1150      | 1150    |
> | Intermediate | 900    | 900       | 3073    |
> | Advanced     | 400    | 821       | 1255    |
> | Expert       | 1000   | 2000      | 1600    |
> | **Total**    | 3450   | 4871      | 7078    |
>
> ### **Reply to Weakness 2**
> **Answer Accuracy:** On the TV-Online evaluation set, we ensure answer accuracy through a  two-stage verification. During evaluation, we first identify model outputs whose timestamps fall within a predefined time window ($W$) of the ground-truth event. Among these candidates, we select the prediction with the closest timestamp, and then use GPT to judge whether the predicted answer is semantically consistent with the ground truth. To ensure the quality of our data, we conducted a human audit on a random sample of 500 QA pairs. The results show an overall acceptance rate of 94.2%, with an inter-annotator agreement (Fleiss' Kappa) of 0.86, indicating high accuracy and reliability of the generated data. This verification strongly supports the quality of our dataset, and we will include the details in the paper's appendix for reference.
>
> **Difficulty Control:** The difficulty levels are controlled through a structured, task-driven design with four tiers: Basic includes a single trigger and a single answer; Intermediate contains one trigger with multiple valid answers; Advanced includes multiple independent triggers requiring separate responses; and Expert includes multiple interdependent triggers that require cross-trigger consistency. This hierarchical design enables a systematic evaluation from basic perception to complex multi-event reasoning.
>
> The response to Weakness 3 is provided in the following document.

---

> > ### Author Response · Authors · 2025-11-27
> >
> > ### **Reply to Weakness 3**
> > To address the concern about generalization, we evaluated our proposed TOM on two representative proactive video understanding benchmarks: ProactiveVideoQA[2] and Vispeak[1]. For fairness and to control for model capacity, we also trained a larger 7B variant (ToM-7B) under the same settings.
> > We followed official evaluation settings for all benchmarks except ProactiveVideoQA, where official details were unavailable. Consequently, we implemented a fully online evaluation for ProactiveVideoQA: the model reads frames sequentially to make real-time response decisions, and the resulting response trajectory is compared with the standard answers.
> >
> > **ProactiveVideoQA:**  We compared TOM-3B and TOM-7B against strong baselines including VideoLLM-Online and MMDuet. As shown in Table 1, TOM-7B achieves state-of-the-art overall performance (37.35), outperforming MMDuet (34.68) and VideoLLM-Online (23.7). Notably, our model shows substantial advantages in open-domain web videos ([WEB]) and TV shows ([TV]), validating the effectiveness and generalizability of our proposed model and data strategy across diverse online scenarios.
> >
> > **Table 1. Experimental Results on ProactiveVideoQA**
> >
> > |Method|Frames||**WEB**|||**EGO**|||**TV**|||**VAD**||Overall|
> > |:---|:---:|:---:|:---:|:---:|:---:|:---:|:---:|:---:|:---:|:---:|:---:|:---:|:---:|:---:|
> > |||**ω=0**|**ω=0.5**|**ω=1**|**ω=0**|**ω=0.5**|**ω=1**|**ω=0**|**ω=0.5**|**ω=1**|**ω=0**|**ω=0.5**|**ω=1**||
> > |**VideoLLM-Online[7]***|-|25.9|25.9|25.9|25.0|25.0|25.1|17.8|18.3|18.8|25.0|25.0|25.0|23.7|
> > |**MMDuet[9]***|-|37.2|38.9|40.7|**44.0**|**46.0**|**47.9**|20.7|21.1|21.6|**26.4**|**27.4**|**28.5**|34.68|
> > |**ToM-3B**|64|39.2|44.0|48.9|31.1|33.3|35.6|24.4|24.5|24.6|22.3|21.9|21.5|32.65|
> > |**ToM-7B**|64|**40.1**|**45.1**|**50.1**|36.1|39.5|42.9|**27.1**|**28.2**|**29.3**|26.2|26.6|27.1|**37.35**|
> >
> > **ViSpeak-Bench:**  TOM achieves the second-best overall performance as illustrated in Table 2. While the top-ranked ViSpeak (s3) additionally leverages the benchmark’s audio stream, whereas TOM relies only on visual information. The remaining gap is primarily in Anomaly Warning (AW) and Humor Reaction (HR), where cues are often conveyed through speech or acoustic signals that our vision-only model cannot access. In these inherently subjective categories, TOM adopts a more conservative strategy to avoid premature hallucinations. Notably, the ViSpeak paper reports that humans also perform worst on AW and HR, suggesting that the difficulty is intrinsic to the task rather than specific to our approach. This makes TOM’s performance particularly encouraging, as it approaches audio-augmented models using vision alone.
> >
> > **Table 2. Experimental Results on Vispeak**
> >
> > |**Method**|**Params**|**Frames**|**AW**|**VI**|**HR**|**VW**|**VT**|**GU**|**TimeAcc(%)(All)**|**VR**|**AW**|**VI**|**HR**|**VW**|**VT**|**GU**|**TextScore(All)**|**Overall**|
> > |:-----------------------:|:----------:|:----------:|:------:|:------:|:------:|:------:|:-------:|:-------:|:---------------------------:|:------:|:------:|:------:|:------:|:------:|:------:|:------:|:--------------------:|:-----------:|
> > |**FlashVstream**[10]*|7B|1fps|34.00|16.00|48.00|75.00|33.00|99.50|50.92|1.75|1.63|1.31|0.67|4.88|4.61|0.70|2.22|1.24|
> > |**Dispider**[8]*|7B|16|38.50|70.00|44.00|69.00|100.00|99.50|70.17|2.50|1.75|4.06|0.91|0.61|2.49|2.07|2.06|1.63|
> > |**Qwen2.5-VL-7B**[11]|7B|1fps|33.50|77.00|35.00|95.00|85.00|100.00|70.91|2.17|1.97|3.80|0.85|5.00|3.80|1.13|2.75|2.16|
> > |**ViSpeak(s3)**[1]*|7B|1fps|56.50|72.00|83.00|93.00|79.00|99.00|**80.42**|3.75|2.63|3.84|1.07|4.95|3.15|3.36|**3.25**|**2.76**|
> > |**ToM-7B(ours)**|7B|1fps|44.50|**80.00**|50.00|**95.00**|83.00|99.00|$\underline{75.00}$|2.34|1.89|**4.45**|0.86|4.95|3.64|1.25|$\underline{2.76}$|$\underline{2.30}$|
> >
> > We also evaluated TOM on additional streaming video understanding benchmarks, including StreamingBench[5], SVBench[4], OVO-Bench[6], and OmniMMI[3]. TOM achieves the best performance on proactive response tasks and various online multi-turn dialogue tasks across these benchmarks, further demonstrating its strong generalization capability. We will include these results in the revised version.

---

> > > ### Author Response · Authors · 2025-11-27
> > >
> > > ### **References**
> > > [1] Fu, Shenghao, et al. "ViSpeak: Visual Instruction Feedback in Streaming Videos." arXiv preprint arXiv:2503.12769 (2025).
> > >
> > > [2] Wang Y, Meng X, Wang Y, et al. Proactivevideoqa: A comprehensive benchmark evaluating proactive interactions in video large language models. arXiv preprint arXiv:2507.09313, 2025.
> > >
> > > [3] Wang, Yuxuan, et al. "OmniMMI: A Comprehensive Multi-modal Interaction Benchmark in Streaming Video Contexts." Proceedings of the Computer Vision and Pattern Recognition Conference. 2025.
> > >
> > > [4] Yang, Zhenyu, et al. "Svbench: A benchmark with temporal multi-turn dialogues for streaming video understanding." arXiv preprint arXiv:2502.10810 (2025).
> > >
> > > [5] Lin, Junming, et al. "Streamingbench: Assessing the gap for mllms to achieve streaming video understanding." arXiv preprint arXiv:2411.03628 (2024).
> > >
> > > [6] Niu, Junbo, et al. "OVO-Bench: How Far is Your Video-LLMs from Real-World Online Video Understanding?." Proceedings of the Computer Vision and Pattern Recognition Conference. 2025.
> > >
> > > [7] Chen, Joya, et al. "Videollm-online: Online video large language model for streaming video." Proceedings of the IEEE/CVF Conference on Computer Vision and Pattern Recognition. 2024.
> > >
> > > [8] Qian, Rui, et al. "Dispider: Enabling video llms with active real-time interaction via disentangled perception, decision, and reaction." Proceedings of the Computer Vision and Pattern Recognition Conference. 2025.
> > >
> > > [9] Wang, Yueqian, et al. "Videollm knows when to speak: Enhancing time-sensitive video comprehension with video-text duet interaction format." arXiv preprint arXiv:2411.17991 (2024).
> > >
> > > [10] Zhang, Haoji, et al. "Flash-vstream: Memory-based real-time understanding for long video streams." arXiv preprint arXiv:2406.08085 (2024).
> > >
> > > [11] Bai, Shuai, et al. "Qwen2. 5-vl technical report." arXiv preprint arXiv:2502.13923 (2025).

---

### Author Response · Authors · 2025-12-02
**General Response: Summary of Revisions and Extensive New Experiments**

**Dear Area Chair**,

We sincerely thank you for managing the review process. We have carefully addressed the comments from Reviewers `dPUS`, `wxSm`, `eJP8`, and `67ms`. We are encouraged that the reviewers recognized the significance of our work in establishing a new direction for video understanding.

For instance, Reviewer `wxSm` commended the work for tackling a **"highly relevant and forward-looking problem"** and filling a **"much-needed gap"** with the TV-Online benchmark. Reviewer `eJP8` highlighted the **"scale and diversity"** of the dataset and the **"effectiveness"** of the proposed model supported by detailed ablation studies. Similarly, Reviewer `dPUS` acknowledged the large scale of the dataset, while Reviewer `67ms` noted the strengths of our **"agent-style streaming model"** and progressive training pipeline.

Based on the reviewers' valuable suggestions, we have revised the manuscript to incorporate experiments demonstrating the robustness, scalability, and generalization of our framework. Specifically, we report results on a 7B-parameter variant, six additional benchmarks, and rigorous human audits. Below is a summary of how we have addressed the key concerns raised during the review process.

**1. Response to Novelty Concerns**

In response to comments from Reviewers `67ms` and `wxSm` regarding algorithmic novelty, we clarify that the core contribution of this work is infrastructural and conceptual. Rather than introducing isolated algorithmic primitives, we identify and formalize the paradigm shift from passive video analysis to Trigger-Centric Online Video Understanding—a proactive, multi-turn, and asynchronous setting that no existing framework has successfully operationalized. By establishing the TV-Online benchmark, a unified evaluation metric, and the baseline TOM framework, our work provides the necessary foundation to catalyze future algorithmic research in this emerging domain.

**2. Demonstration of Strong Generalization**

To address concerns regarding generalization and model capacity raised by Reviewers `dPUS` and `eJP8`, we trained a ToM-7B variant and extended our evaluation to six established external benchmarks. The results, detailed in the revised **Section 5.5** and **Appendix D.1**, demonstrate robust transferability:
- **SOTA Performance**: ToM-7B achieves state-of-the-art results on ProactiveVideoQA (37.35, surpassing MMDuet), StreamingBench (Overall 48.78), and OVO-Bench (Active Responding 54.68%).
- **Surpassing Commercial Models**: On SVBench, ToM-7B achieves an Overall Score of 5.503, notably outperforming the closed-source commercial model Gemini-1.5-Pro (5.363).
- **Consistency**: We also achieve SOTA results on OmniMMI's online multi-turn subtasks. On ViSpeak, our model secures the second-best performance, competitive with audio-enabled baselines despite being vision-only.

**3. Clarification of Details & Expanded Analysis**

We have addressed the requests for clarity and deeper analysis raised by Reviewers `eJP8`, `wxSm`, and `67ms`:
- **Mechanism Clarity**: In the revised **Section 4.1** and **Appendix B**, we specify that the Fast Inference Mechanism (FIM) utilizes an HSV-based similarity threshold to filter redundant frames, and the Trigger State Management (TSM) module employs a lightweight Qwen3-1.7B model to decouple textual history management.
- **Extended Ablation Studies**: We conducted comprehensive new ablations on inference constraints (varying FPS and token budgets) and training data scaling. The results, presented in `Appendix D.2 and C.5`, confirm that our model remains robust under lower token budgets and higher frame rates, and benefits consistently from data scaling.

**4. Limitations and Future Directions**

Addressing the suggestion from Reviewer 67ms regarding audio-conditioned scenarios and other limitations, we have outlined two key directions in the revised Appendix to further advance the field:

- **Multimodal Trigger Grounding**: To transcend vision-only limitations, we propose extending the protocol with audio-specific tokens (e.g., `<|audio_trigger|>`), enabling the model to ground events in both visual changes and acoustic signals for holistic streaming perception.
- **Adaptive Inference Policies**: Moving beyond static thresholds, we aim to develop learnable skipping policies that dynamically adjust frame sampling rates based on semantic entropy, thereby optimizing the latency-accuracy Pareto frontier.
We believe these revisions and the extensive new experimental evidence comprehensively address the reviewers' concerns and demonstrate the validity and impact of our work.

Once again, we express our sincere gratitude to the Area Chair for your time and dedication in overseeing this review process. We believe the substantial revisions made during this period have significantly strengthened the work, comprehensively addressing the raised concerns and clearly demonstrating the validity, novelty, and broad impact of our framework.

---

### Meta-Review · Area_Chair_6ZCZ · 2026-01-08

**Summary:**

This paper introduces a large-scale benchmark for trigger-centric online video understanding. The authors also proposed a streaming-oriented framework with protocol-level tagging and state management. While the reviewers acknowledged the effectiveness of the framework and the scale of the dataset, major concerns remain regarding the novelty of the technical contributions and the limited scope of the benchmark. The core technical components are incremental adaptations of existing techniques. The performance gap between the proposed online model and offline baselines on TGQA remains substantial, and the justification that this stems from protocol differences does not fully address the underlying gap. Given these concerns about novelty, and the performance gap, the AC recommends rejection.

**Reviewer Concerns:**

Addressed concerns. (1) Generalization evaluation. The authors provided additional experiments on six external benchmarks, showing competitive results. (2) Data quality. The authors show that human acceptance rate on 500 samples is the 94.2%.

Outstanding concerns. (1) Limited algorithmic novelty. The authors' defense that this is an infrastructural contribution does not resolve the novelty concern for a top venue like ICLR. (2) Performance gaps. TOM underperforms offline models on TGQA.

**Reviewer Scores:**

While the rebuttal addressed Reviewer dPUS's concerns about data splits and provided additional benchmark evaluations, the fundamental concerns about the dataset being primarily LLM-synthesized and the limited novelty of the approach remain. I expect this reviewer would maintain their score at 4.

Reviewer 67ms raised the most comprehensive concerns spanning novelty, benchmark design, evaluator bias, and missing modalities. While the authors provided detailed responses, many answers essentially acknowledge limitations and defer to future work. I expect this reviewer would maintain their score at 4.

---

### Decision · Program_Chairs · 2026-01-26

Reject